# Continual Learning with Global Prototypes: Beyond the Scope of Task Supervision

## Abstract

Continual learning aims to sequentially learn from different tasks without catastrophic forgetting. With no assumptions of task dependence, the knowledge learned from observed tasks may not align with that required for future tasks. This may result in models' disruptive updates for learning future tasks, causing abrupt changes to previously learned knowledge (e.g. representation drift [7]) which induces catastrophic forgetting. To reduce such disruptive updates, we connect knowledge for observed and unknown tasks by learning task data representations properly related to a set of global prototypes, which have general-purpose connections and are shared across all tasks. We derive global prototypes and the corresponding objective for NLP tasks. For those tasks, the correlated global prototypes can be obtained from a model pre-trained by masked language modeling. And the data representations that have proper relationships to global prototypes can be learned by specific adaptations of the pre-trained model. We investigate existing adaptation models and propose a neighbor attention model which combines different advantages of existing models for our objective. Experiments show that models learning data representations well related to global prototypes can induce significantly less catastrophic forgetting, without memorizing information from past tasks.

## 1 Introduction

In the continual learning paradigm, models progressively learn a sequence of tasks. This paradigm supports real-world applications which face continuous streams of data and tasks [35, 20]. In practice, models may be under storage constraints to use a fixed structure and under privacy considerations that restrict revisiting of previous tasks' data. These introduce the challenge of *catastrophic forgetting*, where models lose knowledge of previously learned tasks after learning new tasks.

Most prior works address catastrophic forgetting using models that integrate the knowledge of the past and present tasks, i.e. the observed tasks. For example, regularization-based models constrain the deviation of current parameters from the previous ones [27, 56, 2, 29]; replay-based models memorize samples from past tasks and rehearse when learning present tasks [35, 9, 46, 26]. However, since there are no assumptions on task dependence in continual learning, models learned from a set of observed tasks may not contain knowledge needed for unknown *future* tasks [28, 16]. To learn such a future task, these models may have disruptive changes on previously learned knowledge (e.g. representation drift [7]), which still induces catastrophic forgetting. One way to reduce such disruptive updates is to make models consider potential knowledge connections to future tasks.

Our key idea is to build connections between observed and unknown tasks by connecting *task-specific* data representations to a *general-purpose* representation base that is shared across all tasks. In many domains, task-specific information about classes can be represented by specific combinations of general units. For example, consider the data instance 'A boy in a red hooded top is smiling. The

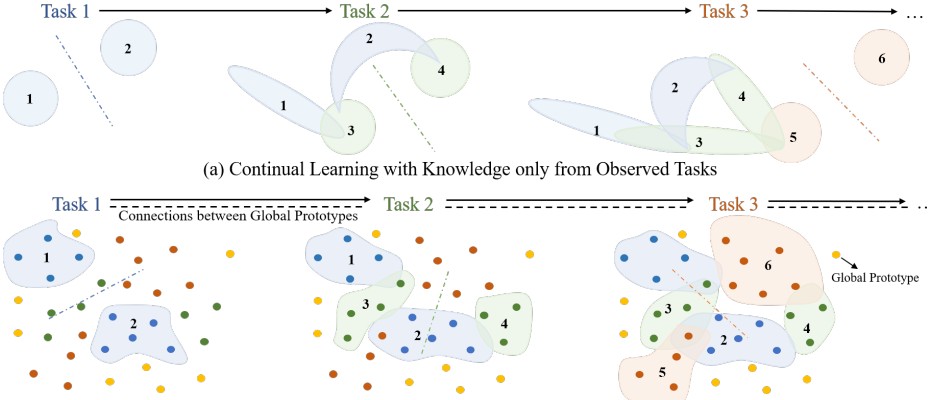

Figure 1: Representations learned with or without global prototypes. The shaded regions cover data representations for each class. In (a), with knowledge only learned for observed supervised tasks, models may have disruptive updates that cause data representation drift when learning a new task. In (b), with reference to correlated global prototypes (dots) in each task learning, representations for different tasks (shaded regions) can properly connect to each other which reduces representation drift.

*boy is upset.*' from '*contradiction*' class in an entailment classification task. The set {*smiling*, *upset*} conveys the task-specific information of '*contradiction*' using the general (i.e. not task-specific) semantics of the token units '*smiling*' and '*upset*'. Based on this, we construct a general-purpose representation base consisting a set of unit representations, which we call *global prototypes*. These global prototypes are pre-learned to reflect semantic connections between them. Then we learn data representations with appropriate task-specific connections to global prototypes. This allows knowledge learned from observed tasks to connect to that of future tasks via the interconnection of global prototypes, which is beyond the scope of task supervision from observed tasks. Our idea mimics mechanism in the brain, a biological continual learning system [56] which rewires existing neurons instead of creating new neurons to learn new tasks [17]. Here, global prototypes mimic the neurons, and learning different connections between data representations and global prototypes mimic the rewiring process. A figure of the idea is shown in Figure 1.

We address two main challenges in realizing this idea: (1). constructing the representation base with correlated global prototypes; (2). learning data representations with task-specific connections to global prototypes. We investigate the above challenges for NLP tasks. For text, the non-contextual token representations are a natural choice for global prototypes, as any text information can be represented by sets of tokens from a fixed vocabulary. For the first challenge, we obtain the global prototypes from a pre-trained language model which learns semantic connections between tokens through self-supervised learning [11]. For the second challenge, we learn data representations by lightly adapting a pre-trained model to obtain task-specific connections to the global prototypes (Section 3). We investigate existing adaptation models with learnable projections (*Adapters* [21]), learnable embeddings (*Prompt Tuning* [30]), and propose a neighbor attention module combining properties of these two (Section 4). Results show that catastrophic forgetting can be significantly mitigated with models that can learn representations well connected to global prototypes. In addition, our neighbor attention model combines the advantages of existing adaptation models, and achieves superior performance in both vanilla and replay settings.

In conclusion, our contributions in this paper are:

1. We propose to learn task-specific information over a general-purpose base with global prototypes to address general task connections in continual learning. Specifically, we derive the construction of the base and the corresponding objective for NLP tasks.

2. We investigate existing adaptation models and propose a new neighbor attention model to learn data representations that have proper relationships to global prototypes.

3. We conduct experiments on different adaptation models and continual learning frameworks. Results show our model can significantly reduce forgetting without replay.

## 2 Related Work

**Continual Learning** Continual learning aims to sequentially learn new tasks while not forgetting previously learned tasks. Models for continual learning can be divided into three main categories: (1). regularization-based models which constrain the deviation of new parameters from the older ones [27, 56, 2, 29]; (2) replay-based models which reduce catastrophic forgetting by rehearsing on real or pseudo samples from previous tasks [35, 9] or generative models [46, 26]; (3). architecture-based models which learn evolving architectures for sequential tasks, with their capacities for each task carefully assigned [44, 53]. Most works above focus on knowledge based on observed tasks.

Some recent works show that knowledge only from observed task supervision is insufficient for continual learning. Knoblauch et al. [28] claim that optimal continual learning requires perfect memory and is NP-hard; Guo et al. [16] suggest preservingholistic information which may not benefit current task but help future tasks. With biased knowledge, models can have disruptive updating when learning a new task, causing problems like representation drift [7, 36, 24]. In this paper, we propose to consider knowledge beyond observed task supervision through a general-purpose base with pre-learned global prototypes. Unlike previous works [39, 4] which use a pre-defined classifier to help class separation [38], our global prototypes are pre-learned with general semantic connections and thus can build connections between tasks. Some works use self-supervised learning to learn more general representations for continual learning [36, 14, 15]. However, those representations do not necessarily connect to specific global prototypes, which is different from our objective.

Continual learning for NLP is an emerging area [3]. Liu et al. [32] introduce a sentence encoder with matrix conceptors; MBPA++ [10] uses an episodic memory with replay and local adaptation to mitigate catastrophic forgetting; LAMOL [49] learns to generate training samples for replay based on pre-trained knowledge; IDBR [23] disentangles hidden spaces to distinguish task-agnostic and task-specific information. Most of them require a memory of past task information, or converting data to a question-answering format along with text-to-text models [6, 42]. Our model does not have such restriction. There are also works [25] focusing on knowledge transfer in continual learning.

**Adaptation Models** In this work, we use adaptation models to learn representations connected to global prototypes. Prior works using pre-trained model with light adaptation for target tasks were originally aimed at parameter efficient tuning. Different methods include adding limited trainable parameters on the frozen transformer layer [21, 40, 18, 22]; or selectively updating existing parameters during training [41, 55]. Recent prompt tuning works [31, 30, 34] learn target tasks by trainable prompt embeddings for generalization purposes as well.

Most closely related work are adaptation models used for continual learning [51, 13, 43]. However, most use the models' parameter efficiency to construct progressive memory. Whether utilizing the pre-trained knowledge can help continual learning, why and how they help remain unexplored. Our approach is based on a fixed model *without* progressive memory of parameters. We use the adaptation model for our desiderata, which also provides a metric to interpret whether the model can benefit continual learning. We believe our work can inspire further utilization of adaptation models for CL.

## 3 Learning over Global Prototypes

We consider the following continual learning setting: the model learns from a sequence of tasks, where each task consists of data $\mathcal{D}_\tau = \{(\mathbf{x}_\tau^{(i)}, y_\tau^{(i)})_{i=1}^{n_\tau}\}$. $\mathbf{x}_\tau$ is the input data and $y_\tau$ is the class label. A task identifier $\tau$ is provided at the training time. We consider two scenarios: *task-incremental* and *class-incremental* learning, where models are task-aware or task-agnostic at the inference time [37]. Without replay, we use the same training objective for both task-incremental and class-incremental learning while evaluating them in different ways.

**Notation** $C_\tau$ represents a set of all classes for each task $\tau$, $C = [C_1, ... C_\tau, ...]$ represents all classes for all tasks. For NLP tasks, $V$ represents the set of tokens with global prototypes in the representation base. $\mathbf{w}^i$ is the $i$-th column of a matrix $\mathbf{w}$. Our main model consists of two components: an encoder $f_\theta$ to generate representation $f_\theta(\mathbf{x}_\tau)$ for each data instance $\mathbf{x}_\tau$; and a classifier with matrix $\mathbf{w}_\gamma \in \mathbb{R}^{d \times |C|}$ for class prediction, where $d$ represents the dimension of data representations. At the inference time, the class label is predicted by $\arg \max_{i \in C_{\text{candidate}}} f_\theta(\mathbf{x}_\tau) \cdot \mathbf{w}_\gamma^i$. For task-incremental inference we have $C_{\text{candidate}} = C_\tau$, while for class-incremental inference we have $C_{\text{candidate}} = C_{1:\tau}$.

## 3.1 The Learning Objective

**Classification Loss** For a task $\tau$, the typical classification objective is to minimize the cross-entropy loss $\mathcal{L}_c(\mathbf{x}_\tau; \theta, \gamma)$ over the training data for the task, as shown below:

$$\mathcal{L}_c(\mathbf{x}_\tau; \theta, \gamma) = -\log \frac{\exp\left(\mathbf{w}_\gamma^{y_\tau} \cdot f_\theta(\mathbf{x}_\tau)\right)}{\sum_{c \in C_\tau} \exp\left(\mathbf{w}_\gamma^c \cdot f_\theta(\mathbf{x}_\tau)\right)}. \tag{1}$$

After learning task $\tau$, models have knowledge about data $\mathbf{x}_{1:\tau}$ and class vectors $\mathbf{w}_\gamma^{c \in C_{1:\tau}}$ from observed tasks $1 : \tau$. However, the knowledge may not align with that required for the unknown future task $(\tau + 1)$. Specifically, after adjusting $\theta$ in task $(\tau + 1)$, the alignment between $\mathbf{w}_\gamma^{y_\tau}$ learned from task $\tau$ and $f_\theta(\mathbf{x}_\tau)$ with adjusted $\theta$ may shift and degrade. In other words, to learn a future task, models may have disruptive updates which make abrupt changes to previously learned knowledge (e.g. representation drift [7]), and induce forgetting.

**Prototype Loss** To mitigate models' disruptive updates, we consider potential connections between observed and unknown tasks. The connection is built by learning task-specific data representations connected to a general-purpose representation base, which is shared across all tasks. The base consists of global token prototypes (denoted $\texttt{proto}[v]$ for token $v$) which reflect semantic connections between them. In particular, we want the data representation $f_\theta(\mathbf{x}_\tau)$ to be connected to the task-relevant global prototypes. Given a reference probability distribution $p(v|\mathbf{x}_\tau, \mathbf{y}_\tau)$ which indicates the strength of connection between data representation and $\texttt{proto}[v]$, we push the data representations towards the prototypes in proportion to their reference probability. Formally, we define the prototype loss as:

$$\mathcal{L}_v(\mathbf{x}_\tau; \theta) = -\sum_{v \in V} p(v|\mathbf{x}_\tau, y_\tau) \log \frac{\exp\left(\texttt{proto}[v] \cdot f_\theta(\mathbf{x}_\tau)\right)}{\sum_{v' \in V} \exp\left(\texttt{proto}[v'] \cdot f_\theta(\mathbf{x}_\tau)\right)}. \tag{2}$$

In Eq.(2), the softmax is calculated over all global prototypes, i.e. $\texttt{proto}[v]$ for any $v \in V$, regardless of task difference. Such calculation is task-agnostic, while the referenced probability $p(v|\mathbf{x}_\tau, \mathbf{y}_\tau)$ gives task-specific guidance for representation learning. By doing this, Eq. (2) learns representations with task-specific connections to global prototypes. Since global prototypes are pre-learned to reflect semantic connections, representations learned by Eq. (2) can connect across tasks via connections of global prototypes. This can reduce abrupt representation change caused by disruptive updating.

The reference probability $p(v|\mathbf{x}_\tau, \mathbf{y}_\tau)$ gives task-specific guidance for representation learning, where tokens with task-specific information of $\mathbf{x}_\tau$ should have high probabilities. Considering both task-specific and holistic information of the data [16, 36], we set $p(v|\mathbf{x}_\tau, y_\tau) = 1/r_\tau$ when $v$ is one of data's $r_\tau$ rationale tokens, i.e. tokens *in the data* that are essential for class prediction [8], otherwise $p(v|\mathbf{x}_\tau, y_\tau) = 0$. Using multiple rationale tokens as task-specific guidance brings extra benefits to the expressiveness of data representations and global prototypes. First, different data representations from the same class have different guidance. Second, a small number of global prototypes can convey rich information when connecting representations to different sets of global prototypes.

**Learning Objective** Based on the above analysis, our learning objective is to learn data representations that can correctly predict class labels (Eq. (1)); and properly connect to global prototypes (Eq. (2)). The optimal parameters $\theta^*, \gamma^*$ for task $\tau$ should satisfy the desiderata below:

- **Task performance.** $\mathcal{L}_c(\mathbf{x}_\tau; \theta^*, \gamma^*) \leq \mathcal{L}_c(\mathbf{x}_\tau; \theta, \gamma)$ for any $\theta \neq \theta^*, \gamma \neq \gamma^*$ (3)
- **Global alignment.** $\mathcal{L}_v(\mathbf{x}_\tau; \theta^*) \leq a_\tau$ (4)

where $a_\tau > 0$ is a threshold value of the prototype loss. Task performance desiderata (Eq. (3)) can be satisfied by optimization on classification loss in Eq. (1). In the rest of this section, we discuss two questions that are necessary for our desiderata: **(1)**. How to get the semantically connected global prototype $\texttt{proto}[v]$ for Eq. (2)? **(2)**. How to get feasible models for the second desiderata in Eq. (4)?

## 3.2 Pre-trained Models for Prototypes and Data Representations

To get correlated global prototypes and learn data representations with reference to them, we utilize a model pre-trained by masked language modeling (MLM). The MLM objective is to predict masked

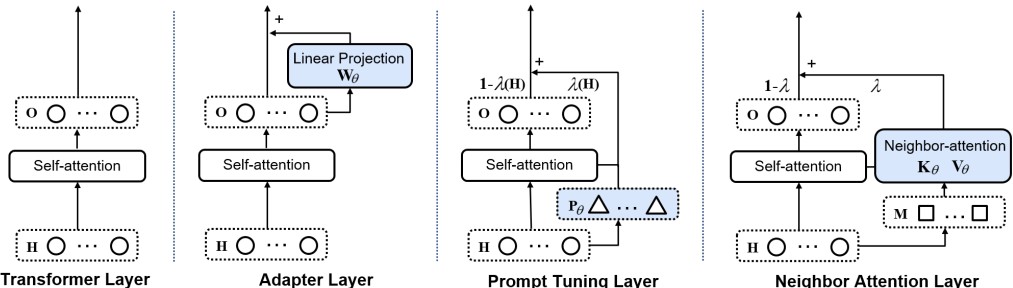

Figure 2: Layers of the transformer and different adaptation models. Shaded blocks are learnable.

token $v_m$ from a masked input $\tilde{\mathbf{x}}$, with the following loss:

$$\mathcal{L}_m(\tilde{\mathbf{x}}; \delta, \phi) = -\sum_{v \in V} p(v|\tilde{\mathbf{x}}) \log \frac{\exp\left(\mathbf{w}_\delta^v \cdot f_\phi(\tilde{\mathbf{x}})\right)}{\sum_{v' \in V} \exp\left(\mathbf{w}_\delta^{v'} \cdot f_\phi(\tilde{\mathbf{x}})\right)}, \tag{5}$$

where $f_\phi$ denotes the encoder for MLM, $\mathbf{w}_\delta$ consists of the token vector $\mathbf{w}_\delta^v$ for each token $v$. The probability $p(v|\tilde{\mathbf{x}}) = 1$ if $v$ is the masked token $v_m$, and 0 otherwise.

**Pre-Trained Model for Global Prototypes**  The MLM objective learns token vectors $\mathbf{w}_\delta$ that reflect semantic connections between tokens, which suits our requirement for global prototypes. Therefore, we can get the global prototype $\texttt{proto}[v]$ as the $v$-th token vector ($\texttt{proto}[v] = \mathbf{w}_\delta^v$) from a model pre-trained by MLM. Extending to cases when pre-trained models are unavailable, we can first train a model by self-supervised learning which learns global prototypes. Global prototypes are fixed once learned. We leave improving them during continual task learning for future study.

**Adapting Pre-Trained Models for Feasibility**  To get feasible models for the desiderata in Eq.(4), we have two options: (a). learning with the prototype loss in Eq.(2); (b). designing a model which can satisfy the desiderata without direct supervision of probabilities $p(v|\mathbf{x}_\tau, y_\tau)$. Option (a) needs rationale tokens to get $p(v|\mathbf{x}_\tau, y_\tau)$, which requires expensive human annotations. In this work, we investigate models for option (b). Specifically, we investigate whether adapting a pre-trained model where we get global prototypes can satisfy our desiderata. Comparing Eq.(5) and Eq.(2), when having $\texttt{proto}[v] = \mathbf{w}_\delta^v$, models for Eq.(5) learn representations that have task-agnostic connections to global prototypes, which is a variant of Eq.(2). When lightly adapting a pre-trained encoder $f_\phi$ to task encoder $f_\theta$, data representations are learned with reference to those task-agnostic connections. Therefore, the adapted representations may have better connections to global prototypes.

In general, our learning includes two stages: first training a model by self-supervised learning for global prototypes (can be skipped if starting from a pre-trained language model); then lightly adapting this model for target tasks while satisfying the desiderata in Eq. (4). We investigate different adaptation models and whether they satisfy our desiderata in the following sections.

## 4  Adaptation Models for Global Alignment

We investigate the potential of different adaptation models for our desiderata of global alignment in Eq.(4). In this section, we first introduce existing adaptation models (Section 4.1) and propose a new neighbor attention model for the desiderata (Section 4.2). A comparison of models is shown in Fig. 2.

### 4.1  Existing Adaptation Models

For a transformer model, representations are calculated by the self-attention mechanism. Given input representations $\mathbf{H} = [\mathbf{h}_1, ..., \mathbf{h}_n]$, each output representation $\mathbf{o}_i$ after self-attention is:

$$\mathbf{o}_i = f\left(\text{MHA}\left(\mathbf{Q}_\phi(\mathbf{h}_i), \mathbf{K}_\phi(\mathbf{H}), \mathbf{V}_\phi(\mathbf{H})\right)\right), \tag{6}$$

where MHA is the multi-head attention function (Appendix A), $f$ is the feed-forward function, $\mathbf{Q}_\phi, \mathbf{K}_\phi, \mathbf{V}_\phi$ are linear functions for query, key and value. Adaptation models utilize pre-trained parameters for self-attentions, while adding extra components to adapt the model for target tasks. According to He et al. [18], different adaptations can be viewed as combining different modification vectors $\Delta_\theta \mathbf{o}_i$ to pre-trained representation $\mathbf{o}_i$. We investigate two types of modifications below.

**Learnable Projections**  Models like *Adapters* [21] insert adaptation modules between transformer layers. The module applies linear projections to the self-attention output $\mathbf{o}_i$, with the non-linear

activation between them. With a residual connection [19], the adapted output $\mathbf{o}_i^{(\text{new})}$ is:

$$\mathbf{o}_i^{(\text{new})} \leftarrow \mathbf{o}_i + \Delta_\theta \mathbf{o}_i, \quad \Delta_\theta \mathbf{o}_i := \mathbf{W}_\theta \mathbf{o}_i. \tag{7}$$

$\mathbf{W}_\theta \in \mathbb{R}^{d \times d}$ represents the linear projections. (We omit the non-linear activation for simplicity).

**Learnable Embeddings**  Models like *Prompt Tuning* [30] add learnable embeddings in the input. Then self-attention is performed based on the input with prompts. The adapted output is [18]:

$$\mathbf{o}_i^{(\text{new})} \leftarrow (1 - \lambda(\mathbf{h}_i))\mathbf{o}_i + \lambda(\mathbf{h}_i)\Delta_\theta \mathbf{o}_i, \quad \Delta_\theta \mathbf{o}_i := \text{MHA}\big(\mathbf{Q}_\phi(\mathbf{h}_i), \mathbf{K}_\phi(\mathbf{P}_\theta), \mathbf{V}_\phi(\mathbf{P}_\theta)\big). \tag{8}$$

$\mathbf{P}_\theta$ are learnable prompt embeddings in $\mathbb{R}^{p \times d}$, $p$ is the number of prompts. $\lambda(\mathbf{h}_i)$ is a gate value computed from self-attention which decides the ratio of pre-trained and modified representations.

**Choices for Global Alignment**  Both of the adaptations show effectiveness in single-task performance for our desiderata Eq. (3) [21, 31]. For global alignment in Eq. (4), *Prompt Tuning* has a gate $\lambda(\mathbf{h}_i)$ to mix pre-trained and modified representations. With a small gate value, this may generate representations close to pre-trained representations, and thus better connect to global prototypes. However, the gate $\lambda(\mathbf{h}_i)$ in Eq. (8) is decided by self attention over inputs and prompts, thus can lean to modified representations $\Delta_\theta \mathbf{o}_i$. Also, the learned prompts $\mathbf{P}_\theta$ may convey information far away from the original data. These may degrade the models' capacity for global alignment. Because of this, we propose a model that has a controlled gate value and relies on neighbors of tokens instead of searching from random prompts for task adaptation. In addition, the training for prompt embeddings is not as easy as that for linear projections [30, 22], which may cause efficiency issues when adapting multiple tasks. We also introduce learnable projections in our model for fast adaptations.

## 4.2 Transformer with Neighbor Attentions

We design a neighbor attention module added to the pre-trained model for task adaptations. The module has three properties: (1). utilizing learnable linear projections to learn modified representations; (2). acquiring neighbor representations for extra information; (3). using a controlled gate to mix pre-trained and modified representations. The adapted output of the neighbor attention module is:

$$\mathbf{o}_i^{(\text{new})} \leftarrow (1 - \lambda)\mathbf{o}_i + \lambda\Delta_\theta \mathbf{o}_i, \ \Delta_\theta \mathbf{o}_i := \text{MHA}\big(\mathbf{Q}_\phi(\mathbf{h}_i), \mathbf{K}_\theta(\mathbf{M}_i || \mathbf{h}_i), \mathbf{V}_\theta(\mathbf{M}_i || \mathbf{h}_i)\big). \tag{9}$$

where $\lambda$ is the ratio of modified representations in the mix-up, $||$ denotes the concatenation operation. $\mathbf{K}_\theta$, $\mathbf{V}_\theta$ are learnable linear functions for key and value. $\mathbf{M}_i = [\mathbf{m}_{i1}, ..., \mathbf{m}_{ik}]$ are $k$ neighbor representations of the input representation $\mathbf{h}_i$.

Comparing Eq. (9) to Eq. (8), neighbor attention has learnable linear functions for key and value. Moreover, we manually control the gate by setting $\lambda = 0.1$ to push the module to focus more on the pre-trained representations. This is for our desiderata to have representations close to pre-trained ones which are trained over global prototypes. Finally, we introduce neighbor representations $\mathbf{M}_i$ for information out of the inputs, which can improve the model's expressivity. Details are shown below.

**Neighbor Representations**  Before the first neighbor attention layer, we find the initial neighbor representations $\mathbf{M}_i$ for a hidden representation $\mathbf{h}_i$. Neighbors of $\mathbf{h}_i$ can be obtained by comparing the dot product between $\mathbf{h}_i$ and token embeddings from the pre-trained embedding layer, then selecting $k$ tokens which have top-$K$ scores as neighbors. $K$ decides the range of the neighborhood.

Then we transform neighbor embeddings to the space of $\mathbf{h}_i$. We disentangle $\mathbf{h}_i$'s $j$-th neighbor representation $\mathbf{m}_{ij}$ into two parts: one related to the hidden representation $\mathbf{h}_i$; and the other related to neighbor information out of $\mathbf{h}_i$. The latter can be obtained by deviating neighbor embedding $\mathbf{e}_{ij}$ from $\mathbf{h}_i$'s token embedding $\mathbf{e}_i$. Then the transformed neighbor representation is: $\mathbf{m}_{ij} = \alpha(\mathbf{e}_{ij} - \mathbf{e}_i) + \beta\mathbf{h}_i$, where $0 < \alpha, \beta < 1$ are scalars. In this paper, we set $\alpha = \beta = 0.2$.

After that, the neighbor representation $\mathbf{M}_i$ is updated at each neighbor attention layer. For the $j$-th neighbor representation $\mathbf{m}_{ij}$, the updated representation $\mathbf{m}_{ij}^{(\text{new})}$ for the next layer is:

$$\mathbf{m}_{ij}^{(\text{new})} \leftarrow \mathbf{m}_{ij} + \Delta_\theta \mathbf{m}_{ij}, \ \Delta_\theta \mathbf{m}_{ij} := f\big(\text{MHA}(\mathbf{Q}_\phi(\mathbf{m}_{ij}), \mathbf{K}_\theta(\mathbf{M}_i || \mathbf{h}_i), \mathbf{V}_\theta(\mathbf{M}_i || \mathbf{h}_i))\big).$$

Adding neighbor attention on more layers will increase the model capacity, but also cause more risk of over-smoothing [45], i.e., neighbor tokens all have the same representations. In practice, we add

neighbor attention to less than half of the transformer layers, and leave the last layer untouched for guidance. In continual learning, the optimal layer selections for different tasks may vary.

## 5 Experimental Settings

**Single Task Evaluation for Desiderata** We first evaluate the models' capacities for our desiderata Eq. (3) and Eq. (4) on single tasks. We test classification accuracies for desiderata of task performance on tasks from the GLUE benchmark [50] and SNLI data [5]. For the desiderata of global alignment, we predict top-20 tokens from the learned representation by the pre-trained decoder (global prototypes), and compute the ratio of rationle tokens in the top-20 predictions (i.e. Recall@20). We evaluate this on e-SNLI dataset [8], where data's rationale tokens [5] are highlighted by human annotators.

**Continual Learning (CL) Evaluation** We evaluate four sequences of tasks: (1) **Yahoo 1**: a split of Yahoo dataset for news question-answer categorization [57] with 5 disjoint tasks containing 2 classes each; (2) **Yahoo 2**: a Yahoo sequence with the same split as (1) but with more data; (3) **DB**: a split of DBPedia data for Wikipedia article classification [57] with 7 disjoint tasks containing 2 classes each; (4) **News Series**: a sequence of tasks on news-related data, including AG_news (news classification, 4 classes), MRPC (paraphrase detection, 2 classes) [12], RTE (text entailment, 2 classes) [52] and SST (sentiment analysis, 2 classes) [47]. For the above sequences except (2), we randomly sample 1245 samples per class, which is the least number of class samples in our datasets. For (2), we sample 10000 samples per class. We measure the average accuracy and forgetting (Appendix C) with standard deviations. For each sequence, we test five random orders of tasks.

We evaluate for both *task-incremental* and *class-incremental* learning. Task identifiers are available at inference time for task-incremental learning but not for class-incremental learning [37]. For class-incremental learning, the original cross-entropy loss over all seen classes will cause significant forgetting [54, 1]. Since our work does not focus on the problem of cross-entropy, we apply the asymmetric strategy (**ACE**) [7]: the current task's classification loss is calculated over in-task classes, while the replay loss is calculated over all seen classes in the memory (if applicable).

**Models and CL Frameworks** We compare different adaptation models on BERT-base. Data representation is from a [MASK] token added to the beginning of input to match the pre-training format. Models for comparison are: (1) **NeiAttn**: our standard neighbor attention model. (2) **NeiReg**: our neighbor attention model with extra regularization for holistic information (Appendix B). (3) **Fine-tuning (FT)**: a model in which all parameters are learnable. (4) **Prompt Tuning (ProT)** [30]: the model adding learnable embeddings only to data inputs. (5) **Prefix Tuning v2 (PT2)** [33]: an adaptation model adding learnable embeddings to inputs of all attention layers. (6) **Adapter** [21]: an adaptation model with learnable linear projections injected in each layer. (7) **BitFit** [55]: an adaptation model tuning only bias terms in the pre-trained model. More settings are in the appendix.

We consider different frameworks (methods) for continual learning: (1) **Vanilla**: the vanilla online learning framework; (2) **MBPA**: an episodic memory framework retrieving stored samples to locally adapt the model at inference time [48]. (3) **ER**: an episodic memory framework storing all seen examples and performs sparse (1%) experience replay; (4) **A-GEM**: an episodic memory framework constraining on gradients to prevent degrading performance of previous tasks [9]; (5) **Probing**: a framework which learns the encoder with Vanilla setting while tunes the classifier for each task using all task data. This is used to evaluate the discrimination of data representations; (6). **MTL**: a muti-task framework that jointly trains on all tasks (not continual learning). For class-incremental cases, we have the above replay-based methods combined with the **ACE** strategy. The baseline performance for each continual learning framework is that on FT model.

## 6 Experimental Results

**Models for Desiderata in Eq.(3) and Eq.(4)** Figure 3 shows models' capacities for our desiderata. We compare the classification accuracy for desiderata in Eq.(3) and Recall@20 of rationale tokens for desiderata in Eq.(4). The higher scores on both metrics, the better model capacities for our desiderata.

Overall, NeiAttn and PT2 consistently achieve a superior balance between classification and recall scores on different NLI tasks. However, Adapter and FT achieve high classification scores but do not generate representations well related to global prototypes (low recall scores). This supports our intuition that mixing pre-trained and modified representations with a gate can result representations better connected to global prototypes. With explicit regularization on holistic information, NeiReg

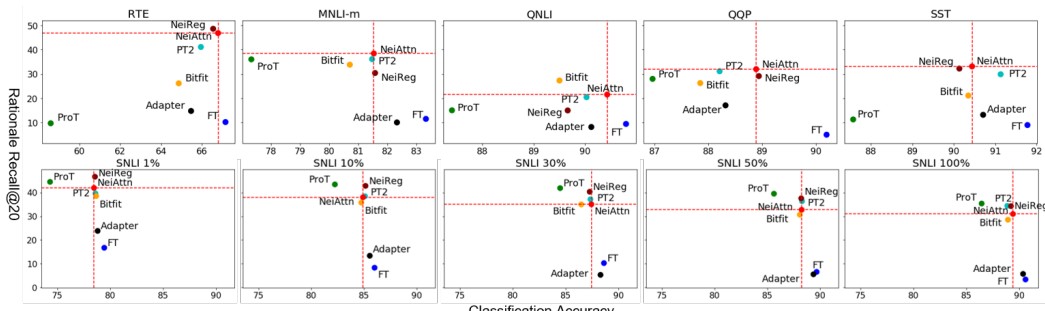

Figure 3: Results for single-task learning. Dashed lines split figure regions based on scores of NeiAttn. Results with higher accuracy and recall (upper right corner) are better. We test on three random seeds.

Table 1: Results for task-incremental learning. We report average accuracy (*Acc*) and forgetting (*Forget*) with their standard deviations (*std*) on five random seeds. **Bold** scores are the best scores and underline scores are the second best. Models in blue have prototype loss larger than the threshold. Models in red satisfy the desiderata Eq. (4). Models with (*) are baselines for each CL framework.

| CL Framework | Model | Yahoo 1 | | Yahoo 2 | | DB | | News Series | |
|---|---|---|---|---|---|---|---|---|---|
| | | *Acc* std | *Forget* std | *Acc* std | *Forget* std | *Acc* std | *Forget* std | *Acc* std | *Forget* std |
| **Vanilla** | Pretrained | 82.95 3.64 | 7.34 3.64 | 83.70 4.16 | 7.71 4.15 | 95.38 2.34 | 4.08 2.37 | 66.66 4.47 | **5.35** 3.06 |
| | FT (*) | 73.07 5.32 | 18.67 5.41 | 79.82 4.29 | 13.27 4.25 | 73.15 5.36 | 24.90 5.17 | 59.98 8.94 | 21.13 7.44 |
| | Adapter | 79.85 1.83 | 11.86 1.83 | 71.90 2.45 | 20.92 2.47 | 98.70 1.10 | 1.19 1.10 | 65.43 4.73 | 15.53 4.29 |
| | PT2 | **88.62** 0.80 | **3.04** 0.79 | **90.64** 0.76 | **2.38** 0.71 | **99.83** 0.04 | **0.07** 0.04 | **75.03** 0.97 | 6.13 0.98 |
| | NeiAttn | **88.96** 1.14 | **2.80** 1.12 | 89.84 0.70 | 3.24 0.69 | 97.34 3.41 | 2.54 3.41 | 71.95 2.20 | 9.89 2.29 |
| **MBPA** | FT (*) | 72.40 4.42 | 19.34 4.49 | 78.71 3.29 | 14.38 3.26 | 73.01 5.45 | 25.04 5.27 | 60.60 8.30 | 20.52 6.67 |
| | Adapter | 78.50 2.12 | 13.13 2.09 | 73.66 2.95 | 19.15 2.95 | 99.09 1.10 | 0.80 1.10 | 65.28 4.74 | 15.67 4.11 |
| | PT2 | **90.69** 0.78 | **0.97** 0.75 | **91.70** 0.51 | **1.33** 0.58 | **99.90** 0.06 | **-0.01** 0.06 | **76.16** 0.81 | 4.99 1.46 |
| | NeiAttn | **90.69** 1.36 | 1.67 1.35 | 91.18 0.90 | 1.90 0.88 | 97.53 3.28 | 2.35 3.28 | 73.28 2.53 | 8.56 2.11 |
| **ER** | FT (*) | 70.77 6.72 | 20.92 6.72 | 90.31 0.72 | 2.67 0.67 | 91.05 8.74 | 8.75 8.69 | 70.44 5.87 | 10.93 4.85 |
| | Adapter | 77.44 3.39 | 14.13 3.42 | 75.79 3.44 | 17.08 3.39 | 98.92 1.54 | 0.97 1.54 | 68.11 1.96 | 13.16 2.10 |
| | PT2 | **88.91** 0.42 | **2.76** 0.35 | 91.02 0.50 | 2.19 0.78 | **99.84** 0.04 | **0.03** 0.03 | 69.60 3.06 | 11.58 2.96 |
| | NeiAttn | 84.02 3.10 | 7.87 3.12 | **91.54** 0.22 | **1.52** 0.24 | 99.68 0.18 | 0.20 0.18 | **75.05** 0.94 | **7.31** 0.48 |
| **A-GEM** | FT (*) | 87.56 1.32 | 4.11 1.40 | 89.98 0.71 | 3.17 0.68 | 84.45 10.16 | 15.34 10.12 | 75.06 6.17 | 5.48 4.01 |
| | Adapter | 80.86 2.36 | 10.65 2.26 | 77.47 3.20 | 15.37 3.24 | 99.52 0.23 | 0.38 0.24 | 73.80 1.16 | 6.72 1.61 |
| | PT2 | 90.40 0.21 | 1.39 0.16 | 90.84 0.19 | 2.22 0.21 | **99.88** 0.01 | **0.01** 0.01 | 73.31 0.73 | 4.29 1.02 |
| | NeiAttn | **90.47** 0.26 | **1.38** 0.21 | **91.35** 0.43 | **1.81** 0.47 | 98.22 3.48 | 1.66 3.49 | **77.07** 1.56 | **4.43** 0.85 |
| **Probing** (classifier non-CL) | FT (*) | 90.18 0.41 | 1.56 0.49 | 92.16 0.14 | 0.93 0.14 | 97.73 3.58 | 0.31 0.04 | 77.17 2.09 | 3.94 1.98 |
| | Adapter | 91.11 0.25 | 0.51 0.25 | 88.98 7.25 | 3.84 7.28 | 99.87 0.01 | 0.02 0.01 | 78.47 0.76 | 2.49 1.70 |
| | PT2 | **91.49** 0.12 | **0.17** 0.09 | **92.81** 0.11 | **0.21** 0.11 | **99.89** 0.01 | **0.01** 0.01 | 77.62 0.32 | 3.53 1.06 |
| | NeiAttn | 91.47 0.16 | 0.29 0.16 | 92.72 0.11 | 0.37 0.11 | 99.87 0.01 | **0.01** 0.02 | **78.83** 0.51 | 3.01 1.02 |
| **MTL** (non-CL) | FT (*) | 91.69 0.26 | — | 92.67 0.71 | — | 99.61 0.41 | — | 79.67 1.99 | — |

performs best in in-task (SNLI→E-SNLI) rationale recalls, while losing its superiority in cross-task (GLUE→E-SNLI) rationale recalls. This may suggest the explicit regularization may not generalize well across tasks. With prompts only in the input, ProT has insufficient capacity for task performance.

For desiderata Eq.(4), NeiAttn and PT2 perform much better than Adapter and FT. We set $a_\tau$ to make NeiAttn and PT2 satisfy Eq.(4) while Adapter and FT fail to, then we evaluate them for CL scenarios.

**Task-Incremental Learning** We test models' capacities for task-incremental learning under different CL frameworks. Results are shown in Table 1. Models are split into two categories according to our desiderata (Eq.(4)) experiment above: (NeiAttn, PT2) which satisfy it and (FT, Adapters) in opposite.

In the vanilla setting, both PT2 and NeiAttn significantly outperform other models with minor forgetting. Adapter on most CL frameworks performs worse than PT2 and NeiAttn, marginally better than FT. This supports our claim that models learning representations better connected to global prototypes perform better in continual learning. Combined with ER and A-GEM, NeiAttn can improve more than PT2 in most cases. FT has significant improvement with replay but can also suffer from overfitting to the replay buffer (ER for Yahoo 1). We also evaluate on a probing framework with only the classifier retrained over task data to evaluate whether the forgetting will cause representations to lose separation. PT2 and NeiAttn also preserve the most separation of representations in this case.

In general, (NeiAttn, PT2) consistently outperform (FT, Adapter) under different CL frameworks. This supports that our desiderata Eq. (4) helps improve models' continual learning ability. NeiAttn performs better with replay. The capacity of models also depends on different data distributions in

the sequence. On News Series, when with replay, FT can even outperform PT2. This may happen because News Series includes data from similar distributions related to the news. And models should have the capacity to deal with knowledge transfer besides catastrophic forgetting.

**Results for Class-Incremental Learning** Figure 4 shows models' performance on class-incremental learning. PT2 and NeiAttn perform well in the vanilla case, where the training is the same as that for task-incremental learning. This indicates that they can address connections between classes from different tasks even without supervision. On the other side, Adapter and FT perform much worse in this case. Then we evaluate three frameworks with replay: one is the full **ER-ACE** [7] with experience replay at each step; one is the **ER-ACE** (sparse) with sparse experience replay; the other is the ACE strategy with only previous task's data stored in the replay (**AGEM-ACE**). We observe that performance on class-incremental learning heavily relies on the quality of replay. In most cases, FT, Adapter and NeiAttn can benefit more from the replay. We hypothesize that it is related to the fast adaptation ability related to linear projections.

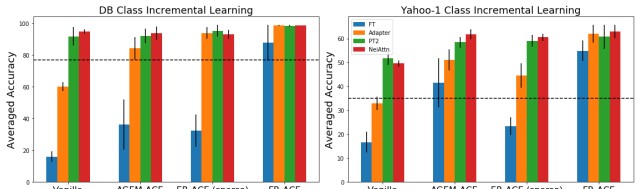

Figure 4: Results on class incremental learning. Dashed lines show scores of a pre-trained model in the vanilla setting.

**Influence of Parameter-Efficiency** With limited parameters, adaptation models have less risk of deviating fast from previously learned knowledge compared to FT, and thus may perform better in CL. However, different models' improvements come not just from having fewer trainable parameters. Table 2 shows the comparison of parameters in each model. NeiAttn has better performance in most cases compared to Adapter and Pre-trained models, which have fewer or no trainable parameters in the encoder. Even with more parameters, NeiAttn performs on par with PT2 with Vanilla and outperform PT2 with replay. NeiAttn also requires much less time to train (5 vs 20 epochs). These suggest the adaptation model structure will highly influence its performance on CL.

Table 2: The ratio of models' learnable parameters compared to FT.

| Models | FT | Bitfit | Adapter | ProT | PT2 | NeiAttn |
|---|---|---|---|---|---|---|
| Parameters (%) | 1 | 0.5 | 2.3 | 0.5 | 0.8 | 4.9 |

**Visualization of Representations** In Figure 5, we visualize NeiAttn and FT's data representations for class-incremental DB under Vanilla and ER-ACE frameworks. Even trained with in-task classes, Vanilla NeiAttn can well disperse data representations. Learning a model includes learning the encoder (representations) and classifier (class vectors). The learned class vectors may not well align with representations even with replay (left bottom). We hypothesize this may result from different training paces for the encoder and classifier. For FT, the encoder quickly learns representations close to single class centroids, which may degrade the function of the classifier. However, with connections to multiple different global prototypes, NeiAttn representations may not quickly move to one centroid. Therefore, it can better balance the training of the encoder and classifier (right bottom).

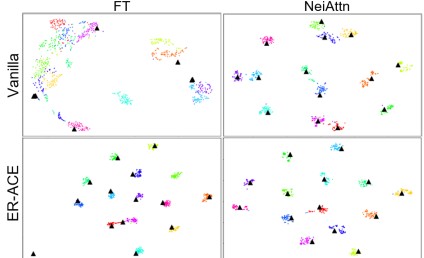

Figure 5: T-SNE plot of FT, NeiAttn representations. Triangles are class vectors.

# 7 Conclusion

In this paper, we investigate models which consider potential connections between observed and unknown tasks to reduce disruptive updating in CL. Specifically, we learn task-specific data representations appropriately connected to a general-purpose representation base with global prototypes. For NLP tasks, the global prototypes can be obtained from a pre-trained language model. And the representation connected to global prototypes can be obtained by lightly adapting the pre-trained model. We investigate existing adaptation models and propose a neighbor attention model which combines advantages of existing models. Experimental results show that models learning representations appropriately connected to global prototypes have significantly less catastrophic forgetting in CL, even without using experience replay. Specifically, when neighbor attention is used, we suffer from less catastrophic forgetting than FT and Adapter, and surpass PT2 when experience replay is applied. We consider the main limitations of our work as: (1) requiring extra memory to compute neighbor attentions; (2) the optimal number of neighbor attention layers may vary for different tasks.

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
