# Appendix

## A   Multi-Head Self-Attention

In single-head self-attention, we compute attention based on queries $\mathbf{Q} \in \mathbb{R}^{p \times d_s}$, keys $\mathbf{K} \in \mathbb{R}^{g \times d_s}$ and values $\mathbf{V} \in \mathbb{R}^{g \times d_s}$:

$$\text{Attn}(\mathbf{Q}, \mathbf{K}, \mathbf{V}) = \text{softmax}(\frac{\mathbf{Q}\mathbf{K}^{\mathbf{T}}}{\sqrt{d_s}})\mathbf{V},$$

where $p$, $g$ are lengths of query and key-value representations, usually we have $p = g$. $d_s$ is the dimension of the query/key/value features. In multi-head attentions, we define the query, key, value functions of the $i$-th head for a sequence of representations $\mathbf{X}$ as:

$$\mathbf{Q}_i(\mathbf{X}) = \mathbf{X}\mathbf{W}_q^{(i)}$$
$$\mathbf{K}_i(\mathbf{X}) = \mathbf{X}\mathbf{W}_k^{(i)}, \ \mathbf{V}_i(\mathbf{X}) = \mathbf{X}\mathbf{W}_v^{(i)}$$

where $\mathbf{X} \in \mathbb{R}^{p \times d}$ is a sequence of $p$ hidden representations with the dim $d$, $\mathbf{W}_q^{(i)}, \mathbf{W}_k^{(i)}, \mathbf{W}_v^{(i)} \in \mathbb{R}^{d \times d_s}$ are projection matrices of queries, keys and values, respectively. The output of the multi-head attention is a concatenation of outputs of all heads:

$$\text{MHA}(\mathbf{Q}(\mathbf{X}), \mathbf{K}(\mathbf{X}), \mathbf{V}(\mathbf{X})) = \text{cat}(\text{head}_1, ..., \text{head}_t)$$
$$\text{head}_i = \text{Attn}(\mathbf{Q}_i(\mathbf{X}), \mathbf{K}_i(\mathbf{X}), \mathbf{V}_i(\mathbf{X}))$$

where $t$ is the number of heads, and typically we have $d = td_s$. The output is then projected by a linear function $f : \mathbb{R}^d \to \mathbb{R}^d$, and followed by the layer normalization and residual connection to get the final output of the layer.

## B   NeiReg: Explicit Regularization For Holistic Information

One property of the neighbor attention model is that neighbor representations correspond to each input token. This makes it easier to guide the training of those representations than arbitrarily prompts. Based on this, we further guide data representations to contain holistic information about the data, which may benefit continual learning [3].

**Context Regularization**   To encourage data representations to contain holistic information of tokens in the data, we encourage sentence (data) representation $\mathbf{o}_{\texttt{[MASK]}}$ at each layer to be close to each token's neighbor representations. Specifically, we maximize the cosine similarity between them, which can be written as the loss:

$$L_{\text{context}}(\mathbf{o}^{(\text{new})}) = -\mathbb{E}_{ij}\big[\cos\big(\mathbf{o}_{\texttt{[MASK]}}^{(\text{new})}, sg(\mathbf{m}_{ij}^{(\text{new})})\big)\big],$$

where $sg$ means 'stop gradient'. We estimate the expectation on neighbors by uniformly sampling a neighbor from the $k$ neighbors of $\mathbf{h}_i$ at each time, when calculating the cosine similarity.

**Neighborhood Regularization**   The context regularization above is directly influenced by neighbor representations. If neighbor representations are too far away from original token representations, the loss $L_{\text{context}}$ may not encourage the model to learn holistic information about data. To mitigate this problem, we add regularization to discourage neighbor representations from moving too far away from token representations. The regularization term is:

$$L_{\text{neigh}}(\mathbf{M}^{(\text{new})}) = -\mathbb{E}_{ij}\cos\big(sg(\mathbf{o}_i^{(\text{new})}), \mathbf{m}_{ij}^{(\text{new})}\big).$$

29  This encourages each neighborhood $\mathbf{M}_i = [\mathbf{m}_{i1}, ...\mathbf{m}_{ik}]$ to stay nearby to its token representation
30  $\mathbf{h}_i$, and not easily migrate to arbitrary spaces.

31  With regularizations above, the overall objective is to minimize the original classification loss with
32  regularization losses $L_{\text{neigh}}(\mathbf{o}^{(\text{new})}) + L_{\text{context}}(\mathbf{M}^{(\text{new})})$.

## C  Evaluation Metrics

34  **Recall@$k$**   Denote the set of rationale tokens of the $i$-th sample as $\text{rel}_i$, the set of top-$k$ tokens
35  predicted from the learned data representation as $\text{pred}_i@k$. The metric Recall@$k$ calculates the
36  proportion of rationale tokens $\text{rel}_i$ that are predicted in $\text{pred}_i@k$, which is defined as:

$$\text{Recall}@k = \mathbb{E}_i\Big[\frac{|\text{pred}_i@k \cap \text{rel}_i|}{|\text{rel}_i|}\Big].$$

37  Because each data instance in E-SNLI has 5-10 rationale tokens, we use Recall@20 for evaluation.

38  **Average Accuracy and Forgetting**   We use the average accuracy and average forgetting similar in
39  [1] to evaluate the performance in CL scenarios. The specific definitions are described below.

- **Average Accuracy ($Acc \in$ [0,1])**: Let $a_{i,j}$ be the performance of the model on the test set
  of task $j$ after the model is trained on task $i$. The average accuracy after training on all task
  $T$ is:

$$Acc_T = \frac{1}{T}\sum_{j=1}^{T} a_{T,j}.$$

  In this paper, we select $T$ as the end of CL task sequence.

- **Average Forgetting ($Forget \in$ [-1,1])**: Denote $f_{i,j}$ as the forgetting on task $j$ after the
  model is trained on task $i$. $f_{i,j}$ is calculated by:

$$f_{i,j} = \max_{l \in \{1,...,i-1\}} a_{l,j} - a_{i,j}.$$

  And the forgetting after training on the task $T$ is:

$$Forget_T = \frac{1}{T}\sum_{j=1}^{T-1} f_{T,j}.$$

  Our forgetting is slightly different from that in [1] by dividing the number $T$ of all tasks
  instead of $T - 1$. We do this to make the above metrics also indicate models' capacities on
  single tasks, i.e. single-task capacity $\approx Acc_T + Forget_T$.

## D  Examples of Representations with Global Prototypes

51  In this section, we provide examples to show: (1). our idea on NLP tasks; (2). connections between
52  learned data representations and global prototypes; (3). neighbors of hidden representations.

### D.1  An NLP Example of Representations Learned with Global Prototypes

54  Figure 6 provides an example of representation learned with global prototypes in NLP tasks, which is
55  a special case of the main paper Figure 1.

56  For NLP data, their task-specific information can be described by tokens in the data which are
57  essential for task predictions, i.e. rationale tokens. For example, for the data '*A boy in a red hooded*
58  *top is smiling. The boy is upset.*' from '*contradiction*' class, tokens in the set {*smiling*, *upset*} are
59  rationale tokens that convey the information of '*contradiction*'. After learning the global prototypes
60  of tokens, we learn data representations that have strong connections to prototypes of rationale
61  tokens. Because global prototypes are pre-learned to reflect (task-agnostic) semantic connections,

62 representations learned from different tasks can connect to each other via the interconnection of global prototypes.

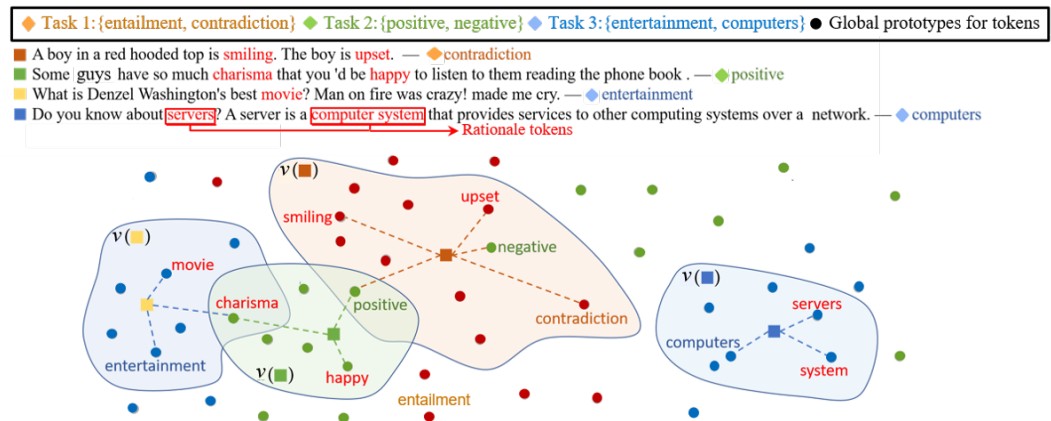

Figure 6: Representations learned with global prototypes for NLP tasks. Shaded regions show ranges of representations for specific classes. Dots are global prototypes, with different colors showing their correlations to specific tasks. The global prototypes are pre-learned to reflect (task-agnostic) semantic connections between them, which may already contain information for specific task learning. Dashed lines show data representation's connection to global prototypes. Specifically, the connection should be strong to global prototypes of corresponding rationale tokens.

63

## D.2 Tokens predicted from Data Representations

65 To evaluate models' capacities for our desiderata of global alignment (the main paper Eq. (4)), we
66 predict top-20 tokens from the learned representation by the pre-trained decoder (global prototypes),
67 and compute the ratio of rationale tokens in the top-20 predictions (i.e. Recall@20). Here we provide
68 examples of top-10 predicted tokens in Table 1. For continual learning data (Yahoo 1 and DB), we
69 show examples from the first task using the model trained after the whole task sequence.

Table 1: Examples of top-10 predicted tokens after training on SNLI (single task), Yahoo 1 (CL) and DB (CL). '*Class*' shows the class text (we use class logits in the model); '*Rationale*' shows essential words annotated in E-SNLI for SNLI data. Underline tokens are rationale tokens in predictions.

| | | |
|---|---|---|
| **SNLI** | *Sentence* 1 | Two women are embracing while holding to go packages. |
| | *Sentence* 2 | The sisters are hugging goodbye while holding to go packages after just eating lunch. |
| | *Class* | Neutral |
| | *Rationale* | The sisters hugging goodbye after just eating lunch |
| **Top-10** | FT | ##dheim Ulysses ##book ##jay 2013 town bankruptcy Odyssey Napoleon Versailles |
| | Adapter | . One Diet Dinner and So Tonight " Today Morning |
| | PT2 | lunch dinner eating breakfast friends leaving food pizza reunion eat |
| | NeiAttn | lunch dinner new goodbye sweet winners miss friends fruit sister |
| **Yahoo 1** (Task 1) | *Sentence* 1 | Who knows how the planets are currently aligned ? ? |
| | *Sentence* 2 | I know how they are aligned but it's difficult to communicate through Yahoo! Imagine that you are in the northern ecliptic sphere looking down on the solar system. Now imagine that there are points at the edge of the solar system like the face of a clock. Let's put the Earth at 90 degrees from the Sun. Mercury would be at approximately 4:00. |
| | *Class* | Science |
| **Top-10** | FT | email emails download ##mail blog blogs downloads Twitter ##pository http |
| | Adapter | : the . Earth us times our - about and |
| | PT2 | planets stars orbits orbit bodies objects positions galaxies coordinates rotation |
| | NeiAttn | Earth Sun Universe earth astronomy celestial spacecraft Jupiter planets solar |
| **DB** (Task 1) | *Sentence* 1 | No One Man |
| | *Sentence* 2 | No One Man is a 1932 American drama film starring Carole Lombard and Ricardo Cortez and directed by Lloyd Corrigan. It is based on a novel by Rupert Hughes. |
| | *Class* | Film |
| **Top-10** | FT | Publishing publishing Publications Press Editorial imprint publications Books readers the |
| | Adapter | directed Film Movie . ! film Pictures ? won ##m |
| | PT2 | film . Film - : Pictures Story ! , Films |
| | NeiAttn | film production distance set screen release The short Film sets |

NeiAttn and PT2 consistently predict rationale tokens along with tokens related to rationale tokens; Adapter can predict limited tokens related to rationale tokens, while many top-10 predicted tokens are irrelevant to data. FT fails to include rationale tokens in its top-10 predicted tokens for most cases. This shows representations of NeiAttn and PT2 can better connect to prototypes of rationale tokens.

## D.3 Correlation between Hidden Representations and Embeddings

In the neighbor selection, we compare hidden representations with the embeddings of the pre-trained model. Here we give an example to show why we can directly compare these two. After transformations over several layers, we still observe a strong correlation between hidden representation $\mathbf{h}$ and its embedding $\mathbf{e}$ at different layers of BERT-base. We hypothesize that it is partially because of the residual connection [4]. In Table 2, by looking at tokens that have the nearest embeddings to the given hidden representation at different layers, we find that embeddings of closely related tokens can be retrieved from such direct comparison. However, after transformation over several layers, the correlation between token embeddings and hidden representations may be decreased. This may call for better neighbor selection strategies.

In Table 2, the top 10 tokens with nearest embeddings have very close meanings to the original token 'prepare', which may not convey sufficient extra information for task learning. In this case, we randomly select $k$ tokens out of top-$K$ nearest tokens for NeiAttn model, where $K$ controls the range of the neighborhood.

Table 2: Top 10 tokens with nearest embeddings of 'prepare' representations at the 1st, 6th and 12th layers on the pre-trained BERT-base model.

| **Input**: [CLS] A group of people *prepare* hot air balloons for takeoff. [SEP] A group of people are outside. [SEP] | |
| --- | --- |
| **Layer** | **Top 10 Nearest Tokens of 'prepare'** |
| 1 | prepare, prepares, preparing, prepared, preparation,preparations, ready, assemble, organize, ##ilize |
| 6 | [CLS], prepare, [MASK], prepared, prepares,preparing, preparation, preparations, assemble, readiness |
| 12 | [CLS], [MASK], [SEP], preparation, readiness,assemble, runoff, ignition, ##itation, prepare |

# E Detailed Experimental Settings

We provide detailed experimental settings in addition to the main paper Section 5. For **single-task** learning and **task-incremental** learning, we use the model settings below:

- **FT**: we select learning rates from {2e-5, 3e-5, 5e-5} and train 3 epochs for each task.
- **Adapter**: we select learning rates from {5e-5, 1e-4, 1e-3} and train {5, 20} epochs for each task. For all continual learning tasks, we train with the learning rate 5e-5 and 20 epochs.
- **BitFit**: we select learning rates from {5e-4, 1e-3} and train {10, 20} epochs for each task.
- **ProT** and **PT2**: for prompt-based models (ProT and PT2), we have learning rate of 1e-3 and train for {20, 50} epochs for each task. Prompt lengths are selected according to their papers. Specifically, we set prompt length as 100 for ProT and 50 for PT2.
- **NeiAttn**: we select learning rates from {2e-4, 5e-4, 1e-3} and train {5, 8} epochs for each task. Neighbor attentions are added on the 7-11 *th* layers of the pre-trained model. The number of neighbors is $k = 5$, the initial neighborhood range is selected from $K =$ {20,50,100}.

For single-task and task-incremental learning, our classifier contains a pooler in addition to the linear classification layer (i.e. $\mathbf{w}_\gamma$ in Eq. (1)) for prediction, which follows the fine-tuning setting used in [2]. The pooler contains a linear projection layer: $\mathbb{R}^d \to \mathbb{R}^d$ followed by a non-linear Tanh activation.

To reduce the forgetting caused by the classifier and focus on the evaluation of representations in **class-incremental** learning, we pick the best encoder-classifier alignment for each model:

- **FT, PT2**: the same setting as task-incremental learning, no pooler in the classifier.
- **Adapter**: the same setting as task-incremental learning, with the pooler in the classifier.

108     • **NeiAttn**: the same setting as task-incremental learning, no pooler in the classifier. For
109     DB, we only add the neighbor attention module on the 7-*th* layer, which has comparable
110     parameters to PT2.

## F   Additional Experimental Results

### F.1   Ablation Study

We conduct ablation studies on specific settings of our NeiAttn model. We evaluate the influence of
hyperparameters including the number of neighbor attention layers, the ratio of neighbor information
in neighbor representations ($\alpha$, $\beta$), and the range of the initial neighborhood ($K$). We evaluate on
SNLI and task-incremental learning tasks. For each task, we select 1245 samples for each class.
Results are shown in Table 3. The standard model applies neighbor attention to 7-11 $th$ transformer
layers, with hyperparameters $\alpha = \beta = 0.2$ and initial neighborhood $K = 20$.

Table 3: Results for ablation studies. We evaluate on SNLI and three task-incremental learning tasks.

| Ablation | NeiAttn Variant | SNLI | | Yahoo 1 | | DB | | News Series | |
|---|---|---|---|---|---|---|---|---|---|
| | | $Acc_{std}$ | $Recall_{std}$ | $Acc_{std}$ | $Forget_{std}$ | $Acc_{std}$ | $Forget_{std}$ | $Acc_{std}$ | $Forget_{std}$ |
| | Standard model | 78.00 $_{0.52}$ | 38.61 $_{2.37}$ | 88.96 $_{1.14}$ | 2.80 $_{1.12}$ | 97.34 $_{3.41}$ | 2.54 $_{3.41}$ | 71.95 $_{2.20}$ | 9.89 $_{2.29}$ |
| **Number of Layers** | 7-7$th$ layers | 73.06 $_{0.24}$ | 30.97 $_{5.96}$ | 90.41 $_{0.18}$ | 1.55 $_{0.19}$ | 99.36 $_{0.71}$ | 0.53 $_{0.71}$ | 74.25 $_{2.12}$ | 3.51 $_{2.14}$ |
| | 7-9$th$ layers | 76.66 $_{0.76}$ | 38.51 $_{3.12}$ | 89.16 $_{0.44}$ | 2.55 $_{0.41}$ | 99.47 $_{0.52}$ | 0.42 $_{0.52}$ | 72.50 $_{2.86}$ | 9.05 $_{2.85}$ |
| | 1-11$th$ layers | 77.52 $_{0.71}$ | 34.30 $_{1.45}$ | 85.49 $_{5.27}$ | 6.28 $_{5.24}$ | 95.53 $_{5.52}$ | 4.34 $_{5.51}$ | 63.98 $_{12.12}$ | 15.85 $_{10.52}$ |
| **Neighbor Information** | $\alpha = 0, \beta = 1$ | 77.86 $_{0.76}$ | 36.61 $_{2.69}$ | 87.39 $_{0.93}$ | 4.23 $_{0.97}$ | 91.84 $_{5.87}$ | 8.04 $_{5.88}$ | 70.04 $_{3.66}$ | 11.64 $_{3.86}$ |
| | $\alpha = 0.2, \beta = 0.8$ | 77.92 $_{0.86}$ | 38.59 $_{1.40}$ | 88.55 $_{0.53}$ | 3.14 $_{0.53}$ | 95.43 $_{4.32}$ | 4.45 $_{4.32}$ | 69.65 $_{3.95}$ | 12.24 $_{4.41}$ |
| | $\alpha = 0.8, \beta = 0.2$ | 77.67 $_{0.80}$ | 38.14 $_{1.22}$ | 89.30 $_{0.36}$ | 2.33 $_{0.38}$ | 97.52 $_{3.37}$ | 2.36 $_{3.36}$ | 69.52 $_{5.14}$ | 11.89 $_{5.22}$ |
| **Range of Neighborhood** | $K = 5$ | 77.67 $_{0.72}$ | 35.31 $_{5.10}$ | 89.33 $_{0.78}$ | 2.40 $_{0.76}$ | 95.11 $_{5.24}$ | 4.77 $_{5.23}$ | 69.53 $_{3.65}$ | 12.13 $_{3.42}$ |
| | $K = 100$ | 77.74 $_{0.86}$ | 38.38 $_{0.74}$ | 89.18 $_{0.63}$ | 2.46 $_{0.63}$ | 96.87 $_{4.51}$ | 3.02 $_{4.51}$ | 69.47 $_{4.08}$ | 12.43 $_{4.00}$ |
| | $K = 200$ | 77.74 $_{0.99}$ | 39.04 $_{1.21}$ | 88.98 $_{0.44}$ | 2.71 $_{0.42}$ | 98.85 $_{0.73}$ | 1.04 $_{0.73}$ | 69.34 $_{4.32}$ | 12.61 $_{4.26}$ |

**Number of Neighbor Attention Layers**  We evaluate NeiAttn models with fewer neighbor attention
layers (applied on the 7$th$ and 7-9 $th$ layers), and more neighbor attention layers (applied to 1-11
transformer layers). With fewer neighbor attention layers, models have less capacity for SNLI
but perform better on CL tasks. This may be because some CL tasks (Yahoo 1, DB) require less
capacity to learn compared to SNLI. And fewer neighbor attention layers may result in less risk of
deviating model parameters to overfit specific tasks. With more neighbor attention layers, the model
performance for SNLI does not increase, while the performance for CL tasks also drops. The above
results suggest the optimal selection of neighbor attention layers may vary for different tasks.

**Ratio of Neighbor Information**  We utilize neighbor representations in Eq. (9) to improve the
model's expressivity without sacrificing its capacity for global alignment (Eq. (4)). For initial neighbor
representations, we mix information of neighbor embeddings and pre-trained hidden representations
by the ratio $\alpha$ and $\beta$ respectively (Main Paper Section 4.2). We compare different ratios of mixed
information. When $\alpha = 0$, $\beta = 1$, the model simply uses the gate $\lambda$ to combine self-attention
representations and a linear projection of hidden representations (without neighbors), as below:

$$\mathbf{o}_i^{(\text{new})} \leftarrow (1 - \lambda)\mathbf{o}_i + \lambda\Delta_\theta\mathbf{o}_i, \ \Delta_\theta\mathbf{o}_i := \mathbf{W}_\theta\mathbf{h}_i.$$

In Table 3, we observe a general performance drop of the model without neighbors compared to that
with neighbors. This validates the benefit of utilizing neighbors in the model.

When increasing the ratio of neighbor information ($\alpha$), tasks in Yahoo 1 and DB will have improved
performance. As analyzed above, NeiAttn with the standard setting has excess capacity for Yahoo
1 and DB, and can overfit single tasks in the sequence. With more neighbor information, models
may have less risk of overfitting and thus perform better in CL. On the other hand, when the model
does not have excess capacity (e.g., on SNLI and News Series), increasing the neighbor information
does not necessarily benefit the learning. In such cases, we may have to carefully set the $\alpha$ and $\beta$ to
balance the information in neighbor representations.

**Range of the Initial Neighborhood**  We evaluate the influence of the initial neighborhood range
$K$. When $K = 5$ which means the neighborhood contains the most similar neighbors, the model
performs better on Yahoo 1 while worse on other tasks compared to the standard model. When
expanding the initial neighborhood with $K > 20$, the model has relatively robust performance on
most tasks. However, on hard tasks like News Series, expanding the initial neighborhood may result
in more variance in the prediction.

 **F.2   Mean and Standard Deviations of Main Paper Figure 3**

Table 4 and Table 5 show the detailed scores (mean and standard deviation) for single task evaluation on SNLI and GLUE datasets (Main Paper Figure 3). Table 4 shows the SNLI scores for Figure 3 bottom, Table 5 shows the GLUE scores for Figure 3 upper.

Table 4: Classification and regularization performance on different ratios of SNLI dataset. 'A' represents the accuracy for classification, 'R' means the Recall@20.

| SNLI (549k) | 1% | | 10% | | 30% | | 50% | | 100% | |
|---|---|---|---|---|---|---|---|---|---|---|
| | A | R | A | R | A | R | A | R | A | R |
| FT | $79.5_{0.8}$ | $16.7_{6.2}$ | $86.0_{0.1}$ | $8.3_{3.3}$ | $88.6_{0.2}$ | $10.2_{2.3}$ | $89.7_{0.3}$ | $6.5_{3.8}$ | $90.6_{0.6}$ | $3.3_{1.0}$ |
| Adapter | $78.8_{1.4}$ | $23.8_{2.7}$ | $85.6_{0.2}$ | $13.3_{2.6}$ | $88.3_{0.4}$ | $5.3_{2.8}$ | $89.4_{0.4}$ | $5.5_{0.9}$ | $90.3_{0.7}$ | $5.7_{0.9}$ |
| BitFit | $78.7_{0.5}$ | $38.5_{0.9}$ | $84.8_{0.6}$ | $35.7_{1.4}$ | $86.5_{0.0}$ | $34.9_{0.4}$ | $88.1_{0.2}$ | $30.6_{0.8}$ | $88.9_{0.4}$ | $28.5_{1.4}$ |
| ProT | $74.3_{0.6}$ | $44.4_{1.1}$ | $82.3_{0.2}$ | $43.4_{0.9}$ | $84.6_{0.2}$ | $41.8_{1.5}$ | $85.6_{0.1}$ | $39.5_{1.9}$ | $86.4_{0.6}$ | $35.3_{1.4}$ |
| PT2 | $78.6_{0.2}$ | $39.6_{0.0}$ | $85.1_{0.2}$ | $38.4_{0.2}$ | $87.3_{0.1}$ | $37.1_{0.9}$ | $88.3_{0.2}$ | $36.3_{0.3}$ | $88.9_{0.5}$ | $34.3_{1.5}$ |
| NeiAttn | $78.5_{0.6}$ | $41.9_{3.0}$ | $84.9_{0.4}$ | $37.9_{2.1}$ | $87.4_{0.1}$ | $35.0_{1.5}$ | $88.3_{0.1}$ | $32.7_{1.2}$ | $89.4_{0.7}$ | $30.9_{1.8}$ |
| NeiReg | $78.6_{0.5}$ | $46.6_{1.1}$ | $85.2_{0.1}$ | $42.8_{0.8}$ | $87.3_{0.0}$ | $40.3_{0.4}$ | $88.3_{0.1}$ | $37.5_{0.5}$ | $89.2_{0.6}$ | $34.2_{1.8}$ |

Table 5: Detailed results of the first row of Figure 3. Classification and regularization performance on selected GLUE datasets. 'A' represents the accuracy for classification, 'R' means the Recall@20.

| | %params | RTE (2.5k) | | SST-2 (67k) | | QNLI (105k) | | QQP (364k) | | MNLI-m (393k) | |
|---|---|---|---|---|---|---|---|---|---|---|---|
| | | A | R | A | R | A | R | A | R | A | R |
| FT | 100 | $67.1_{2.5}$ | $10.3_{6.0}$ | $91.8_{0.4}$ | $9.0_{2.5}$ | $90.8_{0.3}$ | $9.5_{3.4}$ | $90.2_{0.5}$ | $5.1_{3.9}$ | $83.3_{0.1}$ | $11.6_{3.0}$ |
| Adapter | 2.3% | $65.5_{3.4}$ | $14.8_{5.3}$ | $90.7_{0.7}$ | $13.2_{8.3}$ | $90.1_{0.3}$ | $8.2_{5.5}$ | $88.3_{0.6}$ | $17.1_{4.4}$ | $82.3_{0.6}$ | $10.1_{1.5}$ |
| BitFit | 0.5% | $64.9_{2.1}$ | $26.2_{4.9}$ | $90.4_{0.8}$ | $21.1_{5.7}$ | $89.5_{0.2}$ | $27.3_{1.3}$ | $87.9_{0.1}$ | $26.3_{6.1}$ | $80.7_{0.2}$ | $33.8_{0.7}$ |
| ProT | 0.5% | $58.6_{1.6}$ | $9.8_{3.8}$ | $87.6_{0.8}$ | $11.3_{1.2}$ | $87.4_{0.2}$ | $15.1_{5.8}$ | $87.0_{0.1}$ | $28.0_{4.3}$ | $77.3_{0.5}$ | $36.0_{0.9}$ |
| PT2 | 0.8% | $65.9_{2.6}$ | $41.1_{2.6}$ | $91.1_{0.7}$ | $29.9_{0.7}$ | $90.0_{0.1}$ | $20.5_{1.7}$ | $88.2_{0.1}$ | $31.1_{2.2}$ | $81.5_{0.1}$ | $36.1_{0.7}$ |
| NeiAttn | 5.4% | $66.8_{0.7}$ | $46.8_{1.7}$ | $90.4_{0.5}$ | $33.1_{2.0}$ | $90.4_{0.3}$ | $21.6_{3.9}$ | $88.9_{0.0}$ | $32.0_{2.2}$ | $81.5_{0.1}$ | $38.4_{1.3}$ |
| NeiReg | 5.4% | $66.5_{0.7}$ | $48.6_{1.9}$ | $90.4_{0.5}$ | $32.2_{1.0}$ | $90.4_{0.3}$ | $15.0_{2.5}$ | $88.9_{0.0}$ | $29.2_{3.6}$ | $81.5_{0.1}$ | $30.4_{1.0}$ |