# OpenReview forum: "Continual Learning with Global Prototypes: Beyond the Scope of Task Supervision"
_NeurIPS.cc/2023/Conference — Submitted to NeurIPS 2023_

### Official Review · Reviewer_Bgqo · 2023-06-12

**Soundness:** 2 fair
**Presentation:** 2 fair
**Contribution:** 2 fair
**Rating:** 3
**Confidence:** 4

**Summary:**

This paper studies continual learning in NLP by leveraging global prototypes. The authors attribute the catastrophic forgetting to the disruptive updates caused by the misalignment between the knowledge learned from observed tasks and the knowledge required for future tasks. To tackle this problem, the authors propose NeiAttn which derives global prototypes and learns proper relationship between the prototypes and data representations for each task. Experiments show that models learning data representations well related to global prototypes can induce less catastrophic forgetting and NeiAttn outperforms baselines in both task-incremental and class-incremental learning setting.

**Strengths:**

1. The paper and proposed method are well-motivated. The paper reveals a misalignment between the knowledge learned from observed tasks and the knowledge required for future tasks. This is a core issue in continual learning, especially in the continual learning with pre-trained models. The learning objective introduced in Line 154-156 is clear.
2. The paper considers the property of pre-trained language models (discussed in Line 175-178). Personally, I think this perspective is important as pre-training is very common in building machine learning system but previous literatures in continual learning seldom consider the property of pre-trained models.
3. The experimental results show the effectiveness of NeiAttn.

**Weaknesses:**

1. The writing of the proposed method is not clear. Based on my understanding, the core of the proposed method lies in Equation 9, though there are many related contents in Section 3 and Section 4. It would be better if there is an overview of your proposed method given by pseudo code or illustration; or, it may be clearer to introduce your method in a top-down way.
2. NeiAttn outperforms Prompt Tuning marginally according to Table 1 and Figure 4. Based on the motivation and desiderata introduced by this work, I'm wondering if you could apply the training objective in Equation 3, 4 to the Promt Tuning framework. Will it give you better results and simpler method than the current NeiAttn?
3. In Appendix F, the authors give the ablation results of number of neighbor attention layers. The results show that fewer neighbor attention layers give better CL results. If all layers use Neighbor Attention, the results are very bad. Then what's the benefit of using Neighbor Attention for continual learning?

**Questions:**

1. How does the selection of $\alpha_\tau$ influence the results? The paper only mentions "We set $\alpha_\tau$ to make NeiAttn and PT2 satisfy Eq.(4) while Adapter and FT fail to".


**Limitations:**

The paper mentions some limitations in Section 7.

---

> ### Author Rebuttal · Authors · 2023-08-09
>
> Dear reviewer, thank you for the time and effort you have dedicated to evaluating our work. We are glad that you think our perspective in the paper is important. We address your concerns and questions below.
>
> **1\. Points of our method**
>
> Thanks for the suggestion. We would like to illustrate the flow related to our method:
>
> - In Section 3.1, we propose to learn representations related to global prototypes in addition to the conventional classification objective. This leads to our desiderata in the main paper Section 3.1 (Line 154-160). This novel desiderata is the key contribution in our paper, and we derive specific models to realize this desiderata.
> - In Section 3.2, we show how to get global prototypes from models pre-trained by MLM. Based on this, it is possible to adapt a pre-trained model to realize our desiderata instead of explicitly training for Eq. (4), which requires human-annotated rationale tokens (unavailable for most of the datasets). This is the foundation of Section 4, explaining why we study different adaptation structures for our desiderata.
> - Section 4 studies different adaptation structures. It is intuitive that not all adaptations can satisfy our desiderata, and there may be multiple adaptations to satisfy our desiderata. We study existing adaptation structures, and design structures (Eq. (9)) that may better realize our desiderata.
>
> We conclude our contributions are **(1)** proposing a novel desiderata guided by global prototypes, which can improve CL performance; **(2)** finding that pre-trained models can provide such global prototypes and **(3)** introducing an adaptation structure that satisfies our desiderata. Through the experimental study, we find our proposed NeiAttn and existing PT2 both support our claim in (1) and both of them perform well in CL experiments.  We will make our illustration clearer in the paper.
>
> **2\. Performance of NeiAttn**
>
> We would like to clarify a point that may lead to a misunderstanding here. We do not explicitly train the model using Eq. (4), because the annotated rationales are not available for most of the tasks and data. Instead, we look for adaptation structures to satisfy our desiderata.
> We evaluate different models’ ability to satisfy our desiderata in Main Paper Figure 3. Results show that besides our proposed NeiAttn, another adaptation model PT2 also satisfies our desiderata. And the CL experiments in Main Paper Table 1 and Figure 4 show these two models both perform well. These results validate our claim that the adaptation model satisfying our desiderata can perform well in continual learning.
>
> For the simplicity of these two methods,  Prompt-based methods contain fewer parameters but are not easy to train [1,2]. Our NeiAttn requires less time for training and can be further developed for better efficiency with comparable parameters to PT2 (see details below about NeiAttn-LR).
>
> **3\. Study on layers of NeiAttn used**
>
> **Why we do not use NeiAttn on all layers**: Our neighbor attention is designed under the attention mechanism and can have an over-smoothing issue [3], where all neighbor representations become the same after several layers (Main Paper 242-243). If over-smoothing happens, the neighbor attention block actually serves the same as a regular linear layer, which is more likely to overfit data and deviate from the pre-trained knowledge.
>
> We observe that the over-smoothing of NeiAttn can happen after 4-5 NeiAttn layers. So adding neighbor attention on more layers may lead to worse performance in continual learning. However, when learning harder tasks, we may need models with more capacities/parameters, where adding NeiAttn to more layers may help.
>
> **Reasons to use NeiAttn**:
> - Compared to PT2, NeiAttn has better capacity when incorporating replay-based methods (Main Paper Table 1). Also, it requires less training time (\~5 epochs per task) compared to PT2 (\~20 epochs per task).
>
> - The unstable performance of neighbor attention on all layers can be simply improved with parameter efficiency designs. Specifically, we can use low-rank linear layers instead of the fully connected ones, which are used in Adapters and other parameter efficiency models [2]. This will give more robust performance when adding neighbor attention to all layers and have extra efficiency benefits. We show the results of injecting NeiAttn to all layers with and without low-rank (denoted as NeiAttn (all) and NeiAttn-LR (all)) below:
>
>
> |                 |                  |       Yahoo-1 |                |            DB |                 |     News Series |
> |-----------------|-----------------:|--------------:|---------------:|--------------:|----------------:|----------------:|
> | Model           |              Acc |        Forget |            Acc |        Forget |             Acc |          Forget |
> | NeiAttn-LR(all) | 89.40$\pm$0.76   | 2.54$\pm$0.80 | 99.72$\pm$0.13 | 0.17$\pm$0.15 |  70.63$\pm$7.13 |  10.34$\pm$6.52 |
> | NeiAttn(all)    |   85.49$\pm$5.27 |  6.2$\pm$5.24 | 95.53$\pm$5.52 | 4.34$\pm$5.51 | 63.98$\pm$12.12 | 15.85$\pm$10.52 |
>
>
> The low-rank operations may also benefit NeiAttn in the original settings (NeiAttn-LR). Results are shown in Tables 2 and 3 in the rebuttal PDF.
>
> **4\. Clarification on $a_\tau$**
>
> Since we do not explicitly use Eq. (4) for training, the $a_{\tau}$ is only used in the evaluation, as a threshold to decide whether a model can satisfy the desiderata of global alignment. According to Figure 3, there is a clear gap in global alignment performance between (PT2, NeiAttn) and Adapters. So we select the threshold that can distinguish such performance differences. We will make it clearer in the paper.
>
> **Reference**
>
> [1] He et al., Towards a Unified View of Parameter-Efficient Transfer Learning, ICLR 2022
>
> [2] Hu et al., LoRA: Low-Rank Adaptation of Large Language Models, ICLR 2022
>
> [3] Shi et al., Revisiting Over-smoothing in BERT from the Perspective of Graph, ICLR 22

---

### Official Review · Reviewer_fEnP · 2023-06-25

**Soundness:** 3 good
**Presentation:** 3 good
**Contribution:** 2 fair
**Rating:** 4
**Confidence:** 4

**Summary:**

This paper focuses on continual learning in NLP and introduces a regularization-based method to tackle the problem. The main contributions of this work include global alignment, highlighting general-purpose knowledge across tasks, and neighbor attention, which offers a novel parameter-efficient tuning approach. Experimental results on both task-incremental and class-incremental learning scenarios demonstrate the effectiveness of the proposed method.

**Strengths:**

The paper is well-written with smooth flow. It addresses two popular continual learning (CL) settings effectively. The proposed intuition aligns well with LLM and makes logical sense.






**Weaknesses:**

1. The underlying idea is built upon the assumption that LLM contains general-purpose knowledge and that learning tasks should not deviate too far from it. This idea shares similarities with other regularization methods and may suffer from similar drawbacks, such as potential negative impact on new task performance and insufficiency of soft regularization. From this perspective, it is not clear how this paper addresses the problem in a way that other regularization methods cannot.

2. Table 1 lacks inclusion of several CL NLP baselines, (there is an extensive survey in [1]). Only MBPA, published in 2018, is listed for NLP. It is suggested to consider adding more CL NLP baselines, such as [2] (which you have cited) and other latest work mentioned in the survey.

[1]: Continual Learning of Natural Language Processing Tasks: A Survey. https://arxiv.org/abs/2211.12701
[2]: Achieving forgetting prevention and knowledge 443 transfer in continual learning, NeurIPS 2021


**Questions:**

I think this paper has good potential. I would be willing to revise my score if they address the concerns I have raised.






**Limitations:**

See weakness 1

---

> ### Author Rebuttal · Authors · 2023-08-09
>
> Dear reviewer, we sincerely thank you for thoroughly evaluating our work and raising insightful questions. We address your concerns and questions below.
>
>
> **1\. Our regularization vs. others**
>
> Compared to other regularization methods, we regularize data representations’ deviation from the space of global prototypes, rather than the model parameters’ deviation or other representational deviation between tasks. Specifically,
> - Our model contains a pre-trained LM and trainable adaptation blocks. We do not limit trainable parameter deviations during task learning. This gives flexibility in learning new tasks while keeping referencing the pre-trained model for general knowledge.
> - We also do not constrain the deviation of representations between tasks. Our regularization tries to build task-specific representations related to the shared global prototypes. Specifically, representations of different data should be close to different global prototypes that are related to corresponding task predictions. This allows representations to be adequately diverse across (seen and unseen) tasks while staying connected via global prototypes.
>
> We carefully design the adaptation block to further address the balance between task performance and regularization strength. Specifically,
> - We design a trainable structure that is easy to be adapted for task performance.
> - We control the mix-up of representations from the trainable branch and the pre-trained branch for the regularization effect.
> - We increase the model’s task learning capacity by extra token information, while restricting that information to be within the data’s neighborhood for regularization of global alignment.
>
>
> **2\. More NLP baselines**
>
> Thanks for the suggestions. We add two NLP baselines IDBR [1] and CTR [2] in Table 2 and 3 in the rebuttal PDF. Results show that our NeiAttn has overall better performance than these baselines:
> - IDBR works well for task incremental scenarios while performing worse in class incremental scenarios without replay. It also tends to have more representation drift as shown in Figure 1(b) in the rebuttal PDF.
> - CTR is good at tasks requiring knowledge transfer while tending to have forgetting in tasks with little knowledge to share. And it is not applicable to class incremental scenarios. In comparison, our NeiAttn achieves a better balance among different tasks and scenarios.
>
> **References**
>
> [1] Huang et al., Continual Learning for Text Classification with Information Disentanglement Based Regularization, NAACL 2021
>
> [2] Ke et al., Achieving Forgetting Prevention and Knowledge Transfer in Continual Learning, NeurIPS 2021

---

### Official Review · Reviewer_sed9 · 2023-07-05

**Soundness:** 3 good
**Presentation:** 2 fair
**Contribution:** 3 good
**Rating:** 6
**Confidence:** 4

**Summary:**

The authors address the problem of catastrophic forgetting in continual learning and propose to connect observed and unknown tasks by means of task-specific data representations which can be seen as general-purpose representations useful for a wider range of tasks. To this end, they introduce the notion of global prototypes which can be pre-learned and reflect data semantics. Based on these they learn more task specific representations using an objective that trades off two losses (classification loss and prototype loss) . In experimental verifications of their ideas, they consider NLP tasks and find that catastrophic forgetting can successfully be reduced and their neighbor attention model achieves better performance than previous baselines.

**Strengths:**

The broad ideas of this paper are easy to follow. The overall idea for how to mitigate catastrophic forgetting appears simple, sound, and solid. The idea of considering transformers with neighbor attention, too, is simple yet compelling. Experimental evaluations appear to be rigorous and comprehensive; results reveal favorable characteristics of the proposed framework.

**Weaknesses:**

At points there are concerns regarding technical details. Certain statements appear to be handwavy.  Sentences such as „In practice, we add neighbor attention to less than half of the transformer layers and leave the last layer untouched for guidance.” or „In continual learning, the optimal layer selections for different tasks may vary.“ could need more elaboration. Throughout, several hyperparameters are introduced (K, \alpha, \beta, …) which apparently have then been set in a heuristic manner. The contribution would be stronger if a (mathematical) reason for these choices had been given.

**Questions:**

What does it mean to say „… and leave the last layer untouched for guidance.“ ?
Indeed, the concept of “task-specific guidance” seems to be of importance for this work but is not made precise.
In lines 149-150, the ansatz for the reference probability distribution is given and motivated via references [8], [16] and [36]. This connection does not become clear enough. Where do the rational tokens r_\tau come from? The discussion in lines 173-182 does not clarify this point.
How is parameter K in line 234 chosen? Is it set to 20 (as one may guess from line 250)? If so, what motivates this choice? Do other choices lead to different results?
A similar question applies to the statement “in this paper, we set \alpha = \beta = 0.2” (line 239). Some motivation for this choice would strengthen the paper.


**Limitations:**

The authors openly address limitations of their proposed approach (increased memory requirements and additional need for hyper-parameter tuning). There are no concerns w.r.t. to negative societal impact of this work.

---

> ### Author Rebuttal · Authors · 2023-08-09
>
> Dear reviewer,  we appreciate the time and effort you have dedicated to evaluating our work. It is inspiring to see you find our method interesting. We address your concerns and questions below.
>
> **1\. Elaboration on neighbor attention insertion**
>
> Based on the desiderata proposed in Section 3, all our model designs aim to increase the model’s task learning ability without sacrificing the global alignment. Consider each output $o$ of self-attention as: $o = \sum_{i} \lambda_i f(h_i)$ with $\lambda_i \geq 0$ and $\sum_i\lambda_i = 1$, where $h_i$ is the hidden representation at $i$-th position and $f$ is a linear transformation. It searches task-specific information in a convex hull of $f(h_i)$. To improve the model’s expressivity, we add the neighbor attention module to expand such search space with the extra range of neighborhoods.
>
> However, NeiAttn faces the over-smoothing issue, where all neighbor representations become the same after several neighbor attention layers [1]. If over-smoothing happens, the neighbor attention block actually serves the same as a regular linear layer on the neighbor representation, which is more likely to suffer from overfitting and deviate from the pre-trained knowledge. We observe that the over-smoothing can happen after 4-5 neighbor attention layers, and thus inject neighbor attention only to half of the attention layers.
>
> **2\. Optimal NeiAttn layer selection**
>
> In continual learning, tasks may have different difficulties and require different model capacities. Simple tasks may need very light adaptation. In this case, adding extra adaptation blocks to more transformer layers may increase the risk of deviating from the pre-trained knowledge and losing global alignment. On the other hand, hard tasks may require stronger adaptations, and injecting adaptation blocks to more transformer layers may give better performance. We will add more elaborations in the paper.
>
> **3\. Leave the last layer untouched for guidance**
>
> First, we would like to summarize the overall structure of our model: we fix the pre-trained transformer, and add trainable adaptation blocks on selected transformer layers. After that, a classifier is added for label prediction. Based on this overall structure, we do not add the adaptation block to the last transformer layer because the updating (parameters’ gradients) of it will mostly be influenced by the following classifier, which is not pre-trained and purely learned from tasks. Such updating lacks the guidance of the pre-trained knowledge and is less meaningful to our desiderata.
>
> **4\. The reference probability and rationale tokens**
>
> Sorry for the confusion. In Eq. (2), the prototype loss requires to represent data’s task-specific information related to global prototypes, i.e. at the token level. The original numeric label $y_\tau$ is not related to global prototypes. In this case, we should find specific tokens that contain task information for each data. We may have multiple choices to represent such token-level task information. However, based on the paper [2], which suggests preserving data’s holistic information for future task learning, we select tokens that contain information from the original data, and also relate to task predictions. Rationale tokens are good sources of such information. They are extracted from the data tokens, but are related to task predictions. An example of rationales is shown in Supplementary Material D.2. Table 1.
>
> The rationale tokens can be obtained from human annotators, which are however not available for most datasets. In this paper, we use adaptation models to implicitly achieve the effect of global alignment, instead of explicitly training for Eq. (4). In this case, rationale tokens are not required during our training. But we do use a dataset with annotated rationales for evaluation of the global alignment ability (Main Paper Figure 3).
>
> **5\. Hyperparameter selection**
>
> We set hyperparameters ($K$, $\alpha$, $\beta$, …) to control the initial range of the neighborhood. As described above, we want to increase the model’s expressivity by expanding its search space to the neighborhood of the data. The neighborhood can not be too large, otherwise the learned information may deviate far away from the data; and it can not be too small, which may cause a loss of expressivity.
> - "*How is parameter $K$ in line 234 chosen? What motivates this choice? Do other choices lead to different results?*"
> $K$ is set as 20 for experiments in the main paper. $K$ decides the range of neighbor selections. When retrieving neighbors of tokens, we find that the nearest neighbors usually include many variations of the original token (e.g. ‘prepares’ and ‘prepared’ are nearest neighbors of ‘prepare’), which do not contain much extra information. To make extracted neighbors contain some extra information, we pick 5 neighbors out of $K=20$ nearest neighbors for neighbor attention. We have provided an example of neighbors in Supplementary Material D.3 Table 2.
> - "*Motivation of $\alpha$ and $\beta$*". $\alpha$ and $\beta$ control the initial range of neighbor representations. As analyzed before, it should be on an adequate scale. Those hyperparameters are selected mainly by experiments.
>
> We provide the ablation on the influence of these hyperparameters in Supplementary Material F.1.
>
>
> **References**
>
> [1] Shi et al., Revisiting Over-smoothing in BERT from the Perspective of Graph, ICLR 22
>
> [2] Guo et al., Online continual learning through mutual information maximization, ICML 22

---

> > ### Comment · Reviewer_sed9 · 2023-08-21
> >
> > Thank you for your clarification.

---

### Official Review · Reviewer_Zm6a · 2023-07-06

**Soundness:** 2 fair
**Presentation:** 2 fair
**Contribution:** 2 fair
**Rating:** 5
**Confidence:** 3

**Summary:**

This paper proposes a continual learning method for both task and class incremental learnings by incorporating global prototypes. These global prototypes are derived from a pre-trained masked language model and are used to make connections with task specific prototypes. By maintaining these connections, the proposed method prevents task knowledge from forgetting. The authors additionally introduce a trainable module called AttnNei for multi-head attention. The proposed method is evaluated using several existing CL methods and the performance is compared with different adaptation models on BERT-base.

**Strengths:**

1. The overall approach of using global knowledge and task-specific knowledge to prevent forgetting is interesting.
2. The proposed method is applicable to existing CL methods.

**Weaknesses:**

1. I found it difficult to comprehend how and why Eq.2 establishes a connection between global knowledge and task-specific knowledge, leading to the prevention of forgetting.
2. The experiment results do not present good advantage of the proposed method over existing methods due to the following reasons: i) The baselines are too old to know how much improvement it would make when it’s applied to more recent methods. ii) The experiment only demonstrate improvements for CL methods whose original models do not leverage adapters or prompts. However, there re CL methods that already incorporate prompts or adapters [1, 2]. To provide a broader perspective, the authors should compare their method with these approaches. iii) According to Fig 3 and Tab 1, PT2 appears to be comparable in performance, while requiring fewer resources as indicated in Tab 2

3. This method is only applicable to NLP tasks, given that the global prototypes are obtained from language models. Most continual learning methods are designed to be general and not limited to specific task types such as vision or text. Therefore, a discussion on how to apply this method to non-language tasks should be included.

Misc.
Typos: Rationle in line 251. preservingholistic in line 81.

[1] Wang et al. Learning to Prompt for Continual Learning. CVPR, 2022

[2] Kim et al. A multi-head model for continual learning via out-of-distribution replay. CoLLAs, 2022

**Questions:**

1. NeiReg is proposed, but its performance is not compared in Tab 1 and Fig 4.
2. Does the network size expand for each task in the learning process? If it does, how much does it increase at each task and what’s the final size of the model after learning all the tasks?

**Limitations:**

Refer to Weaknesses and Questions

---

> ### Author Rebuttal · Authors · 2023-08-09
>
> Dear reviewer, thank you for thoroughly evaluating our work and raising insightful questions. We address your concerns and questions below.
>
> **1\. The connection between global knowledge and task-specific knowledge in Eq. (2)**
>
> Sorry for the confusion. In Eq. (2), the global knowledge is provided by proto[$v’$] in the denominator, which is the global prototype of the unit $v’$  (i.e. token) shared across all tasks. proto[$v’$] is pre-learned by masked language modeling (Eq. (5)) to reflect task-general connections among all units $v' \in V$.
>
> The task-specific knowledge is provided by $p(v|x_\tau, y_\tau)$, $v \in V$. Specifically for NLP, we set $p(v|x_\tau, y_\tau)>0$ if $v$ is one of rationale tokens of data $x_\tau$. The rationale tokens are tokens in the data that are essential for task predictions (main paper Line 147-150). Generally speaking, instead of the numeric class label $y_\tau$, we describe the task-specific knowledge over global units $v$ (i.e. in the token level). And we encourage the learned representations to be related to prototypes of those global units. Since the global prototypes (proto[$v’$]) are pre-trained and shared across all tasks, this builds the connection between global knowledge and task-specific knowledge. An illustrative example is provided in Supplementary Material D.1 Figure 6.
>
> **Why it can prevent forgetting**: The original classification loss in Eq. (1) learns knowledge only from task supervisions. Specifically, it is guided by class prototypes ${w}^c_\gamma$ which are learned separately for each task, without considering the connections to future tasks. Thus, the knowledge learned with Eq. (1) may not generalize to future tasks. After updating the model for a new task, the new knowledge can interfere with previous knowledge, causing previous task representations to abruptly change (main paper Figure 1(a)). In Eq. (2), we guide each task’s learning with global prototypes. By doing so, representations learned from seen and unseen tasks can align with each other. This may reduce the abrupt representation change and thus reduce forgetting.
>
>
>
> **2\. SOTA Baselines**
>
> i) “*The baselines are too old*”.  Our method does not use experience replay, information of previous tasks, or dynamic structures. We have already compared with recent adaptation baselines (PT2, 2021) and replay-based baselines (ER-ACE, 2022 [6]). Results have shown the improvement of our method to those baselines. We also add more baselines in Tables 2 and 3 of the rebuttal PDF.
>
> ii) "*CL methods that already incorporate prompts or adapters*". We additionally compare L2P[1], CTR[3] as suggested. Since the suggested model in [2] requires replay and OOD data detection besides the adapter model, we compare to another adapter-based CL model CTR for fair comparison. Results are shown in Tables 2 and 3 in the rebuttal PDF.
>
> With prompts only injected in the inputs, L2P has insufficient capacity in single tasks on Bert-base, which may lead to inferior performance in task-incremental learning. CTR, which is designed especially for knowledge transfer, performs well in News Series while tending to have more forgetting in tasks requiring little knowledge transfer. NeiAttn achieves overall better performance than baselines.
>
> **3\. Parameter efficiency and computing resources**
>
> We would like to point out that fewer parameters in PT2 do not mean fewer computations. Prompt-based methods are hard to train [7]. For PT2, it requires 20 epochs for each single task to converge, while NeiAttn only needs 5 epochs. With additional prompts (~50), PT2 also requires additional computing memory in calculating self-attentions.
>
> Since our paper is not for parameter efficiency, we do not design the model with reduced parameters. However, we can achieve parameter efficiency by simply using low-rank linear layers instead of the fully connected ones, which are generally used in parameter efficiency methods like Adapters. We show the results of low-rank NeiAttn (NeiAttn-LR) in Table 2 and 3 in the rebuttal PDF. NeiAttn-LR has comparable parameters to PT2, while preserving the strong CL performance of NeiAttn.
>
> Last but not least, we want to point out that PT2 also satisfies our desiderata for task performance and global alignment. Its success in CL also supports our key claim of this paper.
>
>
>
> **4\. Broader application of proposed approach**
>
> Thanks for the inspiring question. We believe our main contributions on global alignment and the utilization of adaptation models are general to CV tasks. To apply to CV tasks, the keys are to 1. get the pre-trained model with global prototypes, which can be those trained by generative self-supervised learning like [4]; and 2. find proper level adaptations for global alignment like [5].
>
> In addition, we believe our methods’ applications to NLP can strengthen CL research in the NLP domain. Nowadays, large language models and their adaptation models are shown to have good properties in single-task learning. However, for CL, existing works built on pre-trained models/LLMs mainly take advantage of the parameter efficiency in adaptation models, while we show that the global knowledge contained in LLMs can also benefit CL. This may shed light on more CL methods using pre-trained models.
>
> We will add a discussion to the paper.
>
> **References**
>
> [1] Wang et al. Learning to Prompt for Continual Learning. CVPR 2022
>
> [2] Kim et al., A multi-head model for continual learning via out-of-distribution replay, CoLLAs 2022
>
> [3] Ke et al., Achieving Forgetting Prevention and Knowledge Transfer in Continual Learning, NeurIPS 2021
>
> [4] He et al., Masked Autoencoders Are Scalable Vision Learners, CVPR 2022
>
> [5] Zhang et al., Adding Conditional Control to Text-to-Image Diffusion Models, 2023
>
> [6] Caccia et al., New Insights on Reducing Abrupt Representation Change in Online Continual Learning ICLR 2022
>
> [7] He et al., Towards a Unified View of Parameter-Efficient Transfer Learning, ICLR 2022

---

> > ### Comment · Reviewer_Zm6a · 2023-08-16
> > **Response to the author comments**
> >
> > Thank you for the efforts in addressing my concerns. After reading other reviews, I've decided to raise my score from 4 to 5.

---

### Official Review · Reviewer_tk6N · 2023-07-13

**Soundness:** 4 excellent
**Presentation:** 3 good
**Contribution:** 3 good
**Rating:** 5
**Confidence:** 2

**Summary:**

This paper addresses the continual learning problem by leveraging a concept called global prototypes, which are invariant features that are not altered during task-specific continual learning. The training is thus augmented with an additional alignment loss of the data features to the prototypes. The paper then motivates that adapter-like parameter efficient fine-tuning (PeFT) are prototype aligned. Experimental-wise, the paper conducted studies on different PeFT fine-tuning methods for continual learning and showed that their proposed light-adaption NeiAttn and an existing PeFT method PT2 is significantly better than others.

**Strengths:**

The motivation of the paper is clear and the experiments conducted are extensive. The overall presentation is clear. Figures are nice and illustrative.

**Weaknesses:**

1. The paper is dense and while the authors have tried to illustrate it, I still find some parts less convincing. Mostly, I feel like there is simple argument to summarize Section 3: continual learning should not deviate from previous task learned parameters much. This seems to justify all the later experiments and analysis. The current Section 3 draws an abstract component of global prototype, but only realizes it as fixed model parameters, which I feel is redundant. Maybe I missed a point in the paper though, happy to be corrected.
2. There is a lack of measurement on how an adapter module is closer to the original model (closer to the prototype), the paper arrived at a conclusion that PT2 and their NeiAttn are better, but there doesn't seem to be a strong reason behind.

**Questions:**

1. This might be my misunderstanding of the paper, but I absolutely can't parse sentences 300-301, what do you mean by setting a_tau? Was equation (4) actually used in training, or from my understanding, a concept of global prototype that is desired to achieve, but not explictly used in training?

**Limitations:**

some limitations were discussed about the requirement of hyperparameters for their proposed NeiAttn.

---

> ### Author Rebuttal · Authors · 2023-08-09
>
> Dear reviewer, we thank the time and effort you have dedicated to evaluating our work. We address your concerns and questions below.
>
> **1\. Clarification on Section 3**
>
> We would like to clarify the misunderstanding in the statement “there is a simple argument to summarize Section 3: continual learning should not deviate from previous task learned parameters much”. We highlight our important designs below:
>
> - **The use of the fixed model.** We use the fixed model as a common basis to align with global prototypes, rather than forcing model parameters to not deviate from it. Specifically, the fixed model only contains task-general knowledge, and we have to sufficiently adapt that knowledge for specific tasks by learning additional adaptation blocks (e.g. NeiAttn). The learning of adaptation blocks takes a reference to the task-general knowledge (fixed model), but we do not limit deviation of parameters in the adaptation blocks. This means those parameters can still have (large) updates to learn a task if necessary.
>
> - **Not all models can be our fixed model.** Importantly, not all models can be our fixed model with general knowledge aligned with global prototypes. As analyzed in Section 3.2, we can directly use the pre-trained LM because it is pre-trained via masked language modeling (Eq. (5)), which learns model parameters with global prototypes desired in Eq. (3) (i.e. proto[$v$] = $w_\delta^v$). If a fixed model does not contain such global knowledge, adapting it for tasks may not give the effect of global alignment. For example, if we train a model for Task A from scratch, it only contains knowledge of Task A. Then adapting it to an irrelevant Task B may not align the knowledge of Task A and Task B well.
>
> We also conduct additional experiments to verify that our method is not equal to ‘not deviating from previous task learned parameters’. We test EWC [1] which purely constrains the parameter deviation during task learning and empirically find out that it does not outperform our approach in CL experiments. The EWC results are available in Table 2 in the rebuttal PDF.
>
> **2\. Measurement of closeness to the global prototype**
>
> We clarify that we do not encourage the parameters of the adaptation module to be close to the fixed (‘original’) model. Instead, we would expect that the adapted (i.e. final) models produce **representations** within the space of global prototypes. Specifically, global prototypes are representations of some base units (e.g. tokens in NLP). And representations of different data should be close to different global prototypes that are related to corresponding task predictions. An example is shown in Supplementary Material Figure 6.
>
> Based on that, we measure models’ ability to learn representations related to global prototypes by feeding the learned representations to the pre-trained decoder ($w_\delta$ in Eq. (5), recall that we have proto[$v$] = $w_\delta^v$) and predicting top-20 tokens from the decoder. We then compute the ratio of rationale tokens (i.e. tokens with task-specific information of the data) in the top-20 predictions. This measures how the predicted $\hat{p}(v|x_\tau, y_\tau)$ close to ground truth $p(v|x_\tau, y_\tau)$ in Eq. (2) based on global prototypes.
>
> We evaluate this on E-SNLI dataset, where data's rationale tokens are human annotated (main paper line 247-252). The results are shown in the main paper Figure. 3. Compared to PT2 and NeiAttn, Adapters show overall less scores for global alignment (at the right bottom),  that’s why we conclude that PT2 and NeiAttn have better ability for global alignment. An example of predicted rationales is shown in the original Supplementary Material D.2.
>
>
> **3\. Question on $a_\tau$**
>
> Sorry for the confusion. The value of $a_\tau$ decides how strong the desiderata of global alignment is. We do not use $a_\tau$ in our training. The value of $a_\tau$ is used during evaluation to distinguish different models’ global alignment ability.  We will illustrate it clearer in the paper.
>
> **4\. Question on Eq. (4)**
>
> Eq. (4) is **not** explicitly used in training because annotated rationale tokens (for $p(v|x_\tau, y_\tau$) in Eq. (2)) are not available for most datasets (main paper Line 173 - 177).
>
> **Reference**
>
> [1] Kirkpatrick et al., Overcoming catastrophic forgetting in neural networks, PNAS 2016

---

### Author Rebuttal · Authors · 2023-08-09

We sincerely thank all reviewers for their insightful comments and suggestions. We have added new experimental results in the Rebuttal PDF with **(1)** CL baselines using prompt or adapter structures and **(2)** more CL in NLP baselines. Results show that our proposed methods can still achieve advanced results in the overall CL experiments.

For specific questions, we respond to each reviewer respectively. We list the materials used in our rebuttals below:
- Main Paper: the main paper submitted.
- Supplementary Material: the supplementary PDF submitted with the main paper, including examples and ablations.
- Rebuttal PDF: the one-page PDF attached in this general response including our additional experiments.

Thank you for the time to consider our rebuttals.

---

### Decision · Program_Chairs · 2023-09-21

**Decision:**

Reject

**Comment:**

The idea of a global prototype and the correlations of tasks with that prototype is interesting in the context of continual learning. The paper has some merits but at the same time it has some key limitations -- the presentation being unclear, lacking more insights and issues with the evaluation. Considering this, I am in favor of rejecting it in it current version.